# Efficient Near-Optimal Testing of Community Changes in Balanced Stochastic Block Models

**Aditya Gangrade**
Boston University
gangrade@bu.edu

**Praveen Venkatesh**
Carnegie Mellon University
vpraveen@cmu.edu

**Bobak Nazer**
Boston University
bobak@bu.edu

**Venkatesh Saligrama**
Boston University
srv@bu.edu

## Abstract

We propose and analyze the problems of *community goodness-of-fit and two-sample testing* for stochastic block models (SBM), where changes arise due to modification in community memberships of nodes. Motivated by practical applications, we consider the challenging sparse regime, where expected node degrees are constant, and the inter-community mean degree ($b$) scales proportionally to intra-community mean degree ($a$). Prior work has sharply characterized partial or full community recovery in terms of a "signal-to-noise ratio" (SNR) based on $a$ and $b$. For both problems, we propose computationally-efficient tests that can succeed far beyond the regime where recovery of community membership is even possible. Overall, for large changes, $s \gg \sqrt{n}$, we need only SNR $= O(1)$ whereas a naïve test based on community recovery with $O(s)$ errors requires SNR $= \Theta(\log n)$. Conversely, in the small change regime, $s \ll \sqrt{n}$, via an information theoretic lower bound, we show that, surprisingly, no algorithm can do better than the naïve algorithm that first estimates the community up to $O(s)$ errors and then detects changes. We validate these phenomena numerically on SBMs and on real-world datasets as well as Markov Random Fields where we only observe node data rather than the existence of links.

While community detection and recovery for the stochastic block model (SBM) [Abb18] and, more generally, inference of community structures underlying large-scale network data [GN02; New06; For10] has received significant interest across the machine learning, statistics and information theory literatures, there has been limited work on the important problem of testing changes in community structures. The general problem of testing changes in networks naturally arises in a number of applications such as discovering statistically significant topological changes in gene regulatory networks [Zha+08] or differences in brain networks between healthy and diseased individuals [Bas+08]. Building upon this perspective, we propose testing of differences in the underlying community structure of a network, which can encompass scenarios such as detecting structural changes over time in social networks [AG05; For10], determining whether a set of genes belong to different communities in disease and normal states [JTZ04], and deciding whether there are changes in functional modules, which represent communities, in protein-protein networks [CY06].

Testing structural changes in networks is statistically challenging due to the fact that we may have relatively few independent samples to evaluate combinatorially-many potential changes. In this paper, we propose methods for goodness-of-fit (GoF) testing and two-sample testing (TST) for detecting changes in community memberships under the SBM. The SBM naturally captures the community structures commonly observed in large-scale networks, and serves as a baseline model for more complex networks. Specifically, there are $n$ nodes partitioned into two equal-sized communities, and the

network is observed as a random $n \times n$ adjacency matrix, representing the instantaneous pairwise interactions among individuals in the population. Both intra- and inter-community interactions are allowed. Members within the same community interact with uniform probability $a/n$, while members belonging to different communities with a smaller probability $b/n$. We restrict attention to the commonly-considered and practically-relevant setting of $a/b = \Theta(1)$.

For our testing problems, we assume that the network samples are aligned on $n \gg 1$ vertices, and that the latent communities are either the same, or they differ in at least some $s \ll n$ nodes. We pose the GoF problem as: *Decide whether or not the observed random incidence matrix is an instantiation of a given community structure.* For the TST problem, we ask: *Given two random incidence matrices, decide whether or not their latent community structure is identical.*

**Sparse vs. Dense Graphs.** We focus on scenarios where the observed random incidence matrices are sparse with average node degree bounded by a constant independent of the network size. Within this context we develop minimax optimal methods for GoF and TST in this context. We are motivated by both practical and theoretical concerns. Practically, as observed in [Chu10], realistic graphs such as social networks are sparse (friendships do not grow with network size); in temporal settings, at any given time, only a small subset of interactions are observed; and in other cases ascertaining the presence or absence of each edge in the network being observed is an expensive process, and it makes sense to understand the fundamental limits for when testing is even possible.

From a theoretical standpoint, the sparse setting is challenging due to signal-to-noise ratio (SNR) constraints that do not arise in the dense case. Recovery of the latent community with up to $s$ errors is possible iff $\Lambda \gtrsim \log(n/s)$ [CRV15; ZZ16; FC19], where $\Lambda$ is a SNR parameter that, in the setting $a/b = \Theta(1)$, scales linearly with the mean degree. In particular, for $\Lambda$ of constant order, recovery with sublinear distortion fails. The question of *whether testing is possible when recovery fails* is mathematically intriguing. Further, this is the *only* theoretically interesting setting. Indeed, if testing for $s$ changes requires a graph dense enough to allow recovery with $\sim s$ errors, then one might as well recover these communities and compare them.

**Our Contributions.** We show that optimal tests exhibit a surprising two-phase behavior:

1. For $s \gg \sqrt{n}$, or 'large changes,' we propose computationally-efficient schema for GoF and TST that succeed with $\Lambda = O(1)$ - far below the SNR threshold for recovery. For GoF, this requirement is even weaker - we only need $\Lambda \gtrsim n/s^2$, which vanishes with $n$ since $s \gg \sqrt{n}$. Further, we match these bounds up to constants with information-theoretic lower bounds.
2. In contrast, we show via an information-theoretic lower bound that for $s \ll \sqrt{n}$, or 'small changes,' both testing problems require $\Lambda = \Omega(\log(n))$ for reliable testing. This means that the naïve strategy of recovering communities and comparing them is tight up to constants in this regime.

We complement the above theoretical study by three experiments: the first implements the above tests on synthetic SBMs, and the second on the political blogs dataset - a popular real world dataset for community detection [AG05]. Both of these experiments show excellent agreement with the theoretical predictions. The third experiment casts a wider net, and instead studies the related problem of testing the underlying community structure of a Gaussian Markov Random Field that has precision matrix $I + \gamma G$ for $G$ drawn from an SBM. This experiment explores the more realistic setting where instead of receiving a graph, we obtain observations at each node of a hidden graph, and wish to reason about the underlying structure. Remarkably, a simple adaptation of our procedure for SBMs shows excellent performance for this problem. This indicates that our observations are not restricted to raw SBMs, but may signal a more general phenomenon that merits exploration.

**Related Work.** For work on recovery communities we refer to the survey [Abb18]. However, we explicitly point out the papers [CRV15; ZZ16; FC19], which provide various schemes and necessary conditions that show that the partial recovery problem with distortion $s$ can be solved with vanishing error probability if and only if $\Lambda \gtrsim \log(n/s)$. We further point out the lower bounds of [MNS15; DAM17], which assert that if $\Lambda < 2$, then asymptotically, the best possible distortion for partial recovery (or weak recovery, as it is referred to in this constant SNR regime) is $n/2 - o(n)$. Note that reporting a uniformly random community achieves distortion of $s = n/2 - O(\sqrt{n})$.

Ours is the first work to study GoF and TST where both hypothesized models are SBMs. Nevertheless, both GoF and TST in the context of network data as well as SBMs have been studied. Below we highlight the key differences in modeling assumptions and the ensuing technical implications, which renders much of the prior work inapplicable to our setting.

With regards to GoF, [AV14; VA15] study the problem of detecting if a graph is an unstructured

Erdős-Rényi (ER) graph, or if it has a planted dense subgraph, providing detailed characterizations of the feasiblity regions and statistical phase transitions in this setting. While this work is aligned with ours in the techniques used, the modeled setting and problem there are different (ER vs. planted dense subgraph), and TST is not explored. Particularly, the dense subgraph model and the SBM are qualitatively different, and conclusions from one cannot be transferred to the other directly.

A number of papers, including [Lei16; BS16; Ban+16; GL17] study various techniques and regimes of determining if a graph is a SBM or an unstructured ER graph, and if the former, the number of communities in the model. Of these, [GL17] approach the problem by counting small motifs in the graphs, [Ban+16] propose a simple scan and [Lei16; BS16] propose testing of the number of communities on the basis of the top singular values of the graph.

[Tan+17] study TST of the model parameters in random dot product graphs, and propose the distance between aligned spectral embeddings of the two graphs as a statistic to do so. They use this to test equality against various transformations of the underlying models, and in particular for SBMs, test if the connectivity probabilities $(a/n, b/n)$ are identical or not for two graphs with latent communities that are randomly drawn. [LL18] adapt these tests by considering the same distance, but weighted by the corresponding singular values of one of the graphs, and use this to study two-sample testing of equality of the latent communities in the graphs - as in this paper.

In contrast to the low-rank structure assumptions in the above work, [Gho+17a; Gho+17b; GL18] study two-sample testing of inhomogeneous ER graphs (i.e., ER graphs where each edge may have a distinct probability of existing). Within this setting, they provide a number of statistics based both on estimates of the Frobenius and operator norms of the differences of the expected graph adjacency matrices, as well as those based on motifs such as triangles, and explore the limits of these tests.

A fundamental drawback of these approaches, in our context, is their reliance on singular values, spectral norms and Frobenius norms. Singular embeddings are particularly sensitive to noise, and stable embeddings require significant edge density (particularly when a sublinear number of alterations to the communities are to be tested). Indeed, in this context, we note that, in contrast to our low SNR, sparse setting, [LL18] require both a degree of $n^{1/2-\epsilon}$ and an SNR of $\log(n)$ corresponding to a high SNR, high edge-density regime, where full community recovery is possible.

Similarly, Frobenius and Spectral norms based tests of [GL18; Gho+17a] are not stable enough to test a sublinear number of changes in a low SNR regime. Functionally, this can be seen by the fact that the square-Frobenius norm of the difference of two graphs is equal to the number of edges that appear in one graph but not the other, and for sparse graphs, *most* edges appear in only one of the two graphs. Similarly, arguments about spectral norms rely on concentration of the same for ER graphs, but the best known concentration radius [LLV17] is far too large to allow testing of small differences in sparse graphs. Indeed, for any of the statistics of [GL18] to have power in our setting, the results of the paper require that the expected degree diverges with $n$, and that $\Lambda \gtrsim n/s$, which is exponentially above the SNR required to recover communities up to distortion $s/2$.

# 1 Definitions

**The Stochastic Block Model.** A vector $x \in \{\pm 1\}^n$ is said to be a *balanced community vector* (or partition) if $\sum x_i = 0$. The *stochastic block model* is defined as a random, simple, undirected graph $G$ on $n$ nodes such that all edges are drawn mutually independently given $x$, and

$$P(\{i, j\} \in G|x) = \frac{a+b}{2n} + \frac{a-b}{2n} x_i x_j.$$

Note that we treat $x$ as a deterministic but unknown quantity, and thus, $P(\cdot|x)$ is a slight abuse of notation. The parameters $(a, b)$ may vary with $n$, and we focus on the setting $a, b = O(\log n)$, with emphasis on $O(1)$[1], and $a/b = \Theta(1)$. For technical convenience, we require that $a + b < n/4$.

The *signal-to-noise ratio* (SNR) of an SBM is the quantity $\Lambda := \dfrac{(a-b)^2}{a+b}$, which characterises the recovery problem, as described in earlier discussions.

Note that the partitions $x$ and $-x$ induce the same distribution. Accordingly, the *distortion* between partitions $x$ and $y$ is $d(x, y) := \min(d_{\mathrm{H}}(x, y), d_{\mathrm{H}}(x, -y))$, where $d_{\mathrm{H}}$ is the Hamming distance.

**Minimax Testing Problems.** We formally define two minimax hypothesis testing problems.

**Goodness-of-Fit.** We are given a balanced partition $x_0$ and a parameter $s$. We receive a graph $G \sim P(G|x)$, where $x$ is an unknown balanced partition that is either exactly equal to $x_0$ or differs in at least $s$ places. Our goal is to solve the hypothesis test:

$$H_0 : d(x, x_0) = 0 \qquad \text{vs.} \qquad H_1 : d(x, x_0) \geq s$$

We measure the minimax risk of this problem by

$$R_{\mathrm{GoF}}(n, s, a, b) := \inf_{\phi} \sup_{x_0} \left\{ P(\mathrm{FA}) + \sup_{x} P(\mathrm{MD}(x)) \right\} \tag{1}$$

where $\phi(G)$ outputs either 0 or 1, $P(\mathrm{FA}) := P(\phi(G) = 1 \mid x_0)$, $P(\mathrm{MD}(x)) := P(\phi(G) = 0 \mid x)$, are respectively the false alarm and missed detection probabilities, and the second supremum is over all $x$ such that $d(x, x_0) \geq s$.

**Two-Sample Testing.** We are given a parameter $s$ and two independent graphs $G \sim P(G|x), H \sim P(H|y)$, where $x$ and $y$ are unknown balanced communities satisfying $d(x, y) \in \{0\} \cup [s : n/2]$. The goal is to solve the following (composite null) testing problem:

$$H_0 : d(x, y) = 0 \qquad \text{vs.} \qquad H_1 : d(x, y) \geq s,$$

with the measure of risk

$$R_{\mathrm{TST}}(n, s, a, b) := \inf_{\phi} \sup_{x,y} P\big(\phi(G, H) \neq \mathbf{1}\{x = y\} \mid x, y\big), \tag{2}$$

where $\phi(G, H)$ outputs either 0 or 1 and the supremum is over balanced $x, y$ such that $d(x, y) \in \{0\} \cup [s : n/2]$.

As we vary $n$ and $(s, a, b)$ with $n$ as some functions $(s_n, a_n, b_n)$, the above define a sequence of hypothesis tests. We say that the GoF problem can be solved *reliably* for such a sequence if $R_{\mathrm{GoF}}(n, s_n, a_n, b_n) \to 0$ as $n \nearrow \infty$, and similarly for TST. Below, we will target $O(1/n)$ bounds. For conciseness, we will suppress the dependence of risks on $(n, s, a, b)$, writing just $R_{\mathrm{GoF}}/R_{\mathrm{TST}}$.

**On balance:** The strict balance assumption above can be relaxed to only requiring that both communities are of size linear in $n$, at the cost of weakening some of the constants left implicit in the theorem statements. While the majority of the analysis in the paper will assume exact balance, we briefly discuss unbalanced but linearly sized communities whilst detailing the proofs. Note that since the communities are no longer balanced, the differences between $x$ and $y$ can be 'one-sided' i.e., more nodes can move from, say, $+$ to $-$, than in the other direction. We do not require any control on these other than the total number of changes is at least $s$.

**On constants:** We use $C$ and $c$, and their modifications, as unspecified constants that may change from line to line. While these can be explicitly bounded, we do not expect them to be tight.

## 2 Community Goodness-of-Fit

We begin by stating our main results regarding the *community goodness-of-fit problem*.

**Theorem 1.** *Community goodness-of-fit testing is possible with risk $R_{\mathrm{GoF}} \leq \delta$ if $s\Lambda \geq C \log(2/\delta)$ and $\Lambda \geq C \dfrac{n}{s^2} \log(2/\delta)$ for some constant $C > 0$.*

*Conversely, in order to attain $R_{\mathrm{GoF}} \leq \delta \leq 0.25$, we must have that $s\Lambda \geq C' \log(1/\delta)$ and $\Lambda \geq C' \log\left(1 + \dfrac{n}{s^2}\right)$ for some constant $C' > 0$.*

These bounds reveal the following behavior in terms of large and small changes:

- For large changes ($s \geq n^{1/2+c}$ for some $c > 0$), since $n/s^2 \leq 1$ and $\log(1 + x) \geq x/2$ for $x \leq 1$, the second converse bound behaves as $\Lambda \geq Cn/s^2$, matching the sufficient condition up to a constant.
- For small changes ($s \leq n^{1/2-c}$ for some $c > 0$), since $n/s^2 \sim n^{2c}$, the second converse bound instead behaves as $\Lambda \gtrsim \log n$. In this regime, community recovery up to $s/2$ errors requires

$\Lambda \geq C \log 2n/s = \tilde{C} \log n$. Thus, estimating $x$ from $G$ and comparing it to $x_0$ is optimal up to constants.

- The above indicate a phase transition in the GoF testing problem at $\sigma := \log_n(s) = 1/2$. Consider the thermodynamic limit of $n \nearrow \infty$. For $\sigma < 1/2$, the problem is 'hard' in that the SNR $\Lambda$ is required to diverge to $\infty$, while for $\sigma > 1/2$, the SNR can tend to zero.

*Proof Sketch for the Achievability.* Let us begin with an intuitive development of the test. Since we start with a partition $x_0$ in hand to test, it is natural to look at the edges across and within the cut defined by $x_0$. We thus define the number of edgess *across* and *within* this cut:

$$
\begin{aligned}
N_a^{x_0}(G) &:= |\{(i,j) \in G : x_{0,i} \neq x_{0,j}\}| = \frac{1}{4} x_0^\mathsf{T}(D(G) - G)x_0 \\
N_w^{x_0}(G) &:= |\{(i,j) \in G : x_{0,i} = x_{0,j}\}| = \frac{1}{4} x_0^\mathsf{T}(D(G) + G)x_0
\end{aligned}
\tag{3}
$$

where the latter expressions treat $G$ as an adjacency matrix and $D(G) = \operatorname{diag}(\operatorname{degree}(i))_{i \in [1:n]}$. [2] In the null case, these are respectively $\operatorname{Bin}(n^2/4, b/n)$ and $\operatorname{Bin}(2\binom{n/2}{2}, a/n)$ random variables, while in the alternate case some $s/2 \cdot (n-s)/2$ of each behave like edges of the opposite polarity (i.e. as $b/n$ instead of $a/n$ and vice versa), leading to a excess/deficit of edges of this type. Note that while the 'average signal strength', i.e., the amount by which edges are over- or underrepresented is the same in both cases ($\sim s|a-b|$), the group with the larger null parameter suffers greater fluctuations. Thus, we base our test only on edges of smaller bias. This reduces the SNR by at most a factor of $4$.

We now define the test. $C_1$ below is the constant implicit in Lemma 3 in Appendix A.1.

- If $a > b$, we use the test $N_a^{x_0}(G) \underset{H_0}{\overset{H_1}{\gtrless}} \dfrac{bn}{4} + C_1 \max\left(\sqrt{nb \log(1/\delta)}, \log(1/\delta)\right)$.

- If $b > a$, we use the test $N_w^{x_0}(G) \underset{H_0}{\overset{H_1}{\gtrless}} \dfrac{an}{4} - \dfrac{a}{2} + C_1 \max\left(\sqrt{na \log(1/\delta)}, \log(1/\delta)\right)$.

The risks of these tests can be controlled by separating the null and alternate ranges using Bernstein's inequality. Indeed, the threshold above is just the the expectation plus the concentration radius of the statistic under the null distribution. Let us briefly develop the statistic's behaviour in the alternate - considering only the case $a > b$, we find that under the alternate, $\binom{n-s}{2} + \binom{s}{2}$ of the edges in $N_a^{x_0}$ continue to behave like $\operatorname{Bern}(b/n)$ bits, while the remaining $s(n-s)/2$ edges behave as $\operatorname{Bern}(a/n)$ bits. Thus, the expectation of $N_a^{x_0}$ is increased by an amount greater than $s(n-s)\frac{a-b}{2n} \geq s(a-b)/4$. Next, Bernstein's inequality controls the fluctuations at scale $\sqrt{\max(nb, s(a-b)) \log(2/\delta)}$. The conclusion is straightforward to draw from here, and the proof is carried out in Appendix A.1[3].

*Proof Sketch for the Converse.* The proof is relegated to Appendix A.2, and we discuss the strategy here. The converse proof follows Le Cam's method, which lower bounds the minimax risk by the Bayes risk for conveniently chosen priors - which can be expressed using the TV distance.

To show $\Lambda \gtrsim \log(1 + n/s^2)$, we pick the null $x_0$ to be any balanced community, and choose the uniform prior on communities that are exactly $s$-far from $x_0$ (in fact, we only use a subset of these in order to facilitate easier computations). This is an obvious choice for this setting - we are interested in balanced communities that are at least $s$ far, and choosing a large number of them allows for a greater 'confusion' in the testing problem due to a richer alternate hypothesis. The bound follows by invoking inequalities between TV and $\chi^2$ divergences and a lengthy calculation due to the combinatorial objects involved.

To show $s\Lambda \gtrsim -\log(\delta)$, we again pick the null to be any balanced community, and pick the alternate to be an $s$-far singleton. We then proceed to control $d_{\mathrm{TV}}$ by the Hellinger divergence.

## 3 Two-Sample Testing

We again begin with the main results on *community two-sample testing problem*.

**Theorem 2.** *Assume, for some $\gamma > 0$, $s \geq n^{\frac{1}{2}+\gamma}$. There exist constants $C, C'$ such that if $C' \leq a, b \leq (n/2)^{1/3}$, then two-sample testing of $s$ changes with $R_{\mathrm{TST}} \leq 4/n$ is possible if the SNR satisfies $\Lambda \geq C$.*

*Conversely, for $n \geq 200$, there exist constants $c, c'$ such that if $s < (\frac{1}{2} - c')n$, then two-sample testing of $s$ changes cannot be carried out with $R_{\mathrm{TST}} \leq 1/4$ unless $\Lambda \geq c$.*

**Large Changes.** The above theorem makes an achievability claim for the setting of large changes. Notice that in this regime the stated upper and lower bounds match up to constants. Specifically, if $n^{\frac{1}{2}+\gamma} < s < (\frac{1}{2} - c')n$, two-sample testing can be solved iff $\Lambda \gtrsim 1$. Further, the condition $a, b \gtrsim 1$ is also tight, as it follows from $a/b = \Theta(1)$, and the necessary condition $\Lambda \gtrsim 1$, since $\Lambda \leq a + b$.

This leaves the condition $\max(a, b) \leq (n/2)^{1/3}$, which we suspect is an artifact of the proof technique and conjecture that, even for our proposed test, it can be removed. In any case, observe that this condition is irrelevant in the setting $a, b = O(\log n)$ considered in this paper. Further, if $a/b$ is bounded away from 1, then TST is directly possible when $a, b = \Omega(\log n)$ by recovering the communities and comparing them, demonstrating that this condition is not present in general.

**Small Changes.** We claim that for small changes - $s < n^{\frac{1}{2}-\gamma}$ for some $\gamma > 0$ - the naïve scheme of recovering the communities and comparing them is minimax. To see this, note that that GoF testing is reducible to TST - given a TST scheme of a known risk, one may construct a GoF tester of that risk by feeding the TST algorithm the observed graph and a graph drawn from $P(\cdot|x_0)$. Thus, the lower bounds of Theorem 1 apply to TST, and for $a/b = \Theta(1)$, we find that it is necessary that $s\Lambda = \omega(1)$ and that $\Lambda \gtrsim \log(1 + n/s^2)$ to attain vanishing $R_{\mathrm{TST}}$. For small $s$, the latter lower bound is $\Omega(\log n)$, the claim follows since recovery with up to $s$ errors is possible if $\Lambda \gtrsim \log n$.

**Efficiency.** Finally, we point out that the above bounds can be attained with computationally efficient tests. Further, for large changes, the test can be made agnostic to knowledge of $(a, b)$. Instead, it only requires one to be able to estimate $n(a+b)$ to within an additive error of $\widetilde{O}(\sqrt{n(a+b)})$, which can be done by simply counting the number of edges in the graphs.

*Proof Sketch of the Achievability.* We describe the proposed test, and sketch its risk analysis below, completing the same in Appendix B.1. Recall the definition of $N_w^z$, $N_a^z$ from (3) in §2, and let

$$T^{\hat{x}}(G) := N_w^{\hat{x}}(G) - N_a^{\hat{x}}(G). \tag{4}$$

We show that the routine 'TwoSampleTester' below attains a risk smaller than $4/n$. In words, the test computes a partition $\hat{x}$ for the graph $G$ by using about half the edges in the graph. This is represented in the 'PartialRecovery' step below, for which any such method may be used - concretely, that of [CRV15]. Next, we compute the statistic $T^{\hat{x}}$ above for both the remaining part of the first graph, and

for the second graph. Notice that unlike the GoF statistic, which was only $N_a$, $T^{\hat{x}}$ takes the difference of $N_a$ and $N_w$. This is necessary because the partition $\hat{x}$ derived from partial recovery cannot be very well correlated with the true partition $x$. This means the reduced fluctuations from only considering one part does not apply, and we instead use the whole cut.

---

**Algorithm 1:** TwoSampleTester$(G, H, \delta)$

1: $G_1 \leftarrow$ subsampling of edges of $G$ at rate $1/2$ uniformly at random.
2: $\widetilde{G} \leftarrow G - G_1$.
3: $\widehat{x} \leftarrow$ PartialRecovery$(G_1)$.
4: Compute $T^{\widehat{x}}(\widetilde{G}), T^{\widehat{x}}(H)$.
5: $T \leftarrow |2T^{\widehat{x}}(\widetilde{G}) - T^{\widehat{x}}(H)|$.
6: Return $T \underset{H_0}{\overset{H_1}{\gtrless}} \sqrt{Cn(a+b)\log(6n)}$.

---

Since the edges within communities, and across communities in the graph are (separately) exchangable, the errors made in $\hat{x}$ distribute uniformly over the two communities[4]. This allows us to explicitly control the behaviour of $T$ as defined in the test *provided $\hat{x}$ is non-trivially correlatd with $x$* - i.e., given that it makes $< (1/2 - c)n$ errors for some $c > 0$. The condition $\Lambda \gtrsim 1$ in the theorem arises from this.

A complication in this strategy is that the remaining graph $\widetilde{G}$ in the scheme is not independent of the recovered community $\hat{x}$. This is handled in the analysis by introducing an independent copy of $G$, called $G'$, and arguing that $T^{\hat{x}}(\widetilde{G}) \approx 1/2 T^{\hat{x}}(G')$. This step is the origin of the nuisance condition $\max(a, b) \lesssim n^{1/3}$ in the theorem.

Lastly, we point out that while the above exploits the exact balance by using the description of the error distribution it enables, one can derive the same results (but with weakened constants) even without this assumption, so long as both communities are at of size linear in $n$. In this case, one cannot rely on the errors distributing uniformly over the nodes, but the within-community uniformity of errors, which follows due to within community exchangability, can be exploited in a similar way. We describe this extension in Appendix B.1.1.

*Proof Sketch of the Converse.* The necessary condition is shown via Le Cam's method, but with the twist that the null model is chosen to be a two-step procedure - one that draws a balanced community uniformly at random, and then generates a graph according to it, while the alternate models are drawn uniformly from the balanced communities that are at least $s$-far from the chosen null. This allows a comparison to the unstructured Erdős-Rényi graph on $n$ vertices with mean degree $(a + b)/2$. Bounds can then be drawn in from the study of the so-called distinguishability problem [Ban+16], and we invoke results from [WX18] to show that total variation distance between the null and alternate distributions is small when $\Lambda$ is a small enough constant, allowing us to conclude using Neyman-Pearson. See Appendix B.3 for a detailed argument.

## 4 Experiments

We perform three different sets of numerical experiments. We first run our tests on SBMs with 1000 nodes. Next, we demonstrate that our tests perform similarly for a real dataset, specifically the Political Blogs dataset [AG05]. Finally, we examine SBM-supported Gaussian Markov Random Fields (GMRFs) as an example of a "node observation" model, where the SBM-generated edges form the precision matrix for the Gaussian vector consisting of the random variables assigned to each node. In particular, we need to determine if the underlying community of the graph has changed without explicitly observing (or recovering) the edges of the graph. For the sake of brevity, precise details of the experiments are moved to Appendix C.

### 4.1 SBM Experiments

We perform experiments implementing our GoF and TST strategies as well as the naïve scheme of reconstructing communities and comparing. Recovery is performed by regularised spectral clustering, for which a detailed description is given in Appendix C.1. The graphs are drawn on $n = 1000$ nodes for a range of $(s, \Lambda)$ pairs and the high and low risk regimes are plotted in Figure 1. First, note that for 'large changes,' $s \geq \sqrt{n \log(10)} \approx 50$, our GoF and TST tests can succeed for lower SNR values. In contrast, for 'small changes,' $s < \sqrt{n} \approx 30$, the naïve test is more powerful in the high SNR regime. Additionally, both tests fail for TST unless the SNR is larger than a constant, as predicted by our lower bound in Theorem 2.

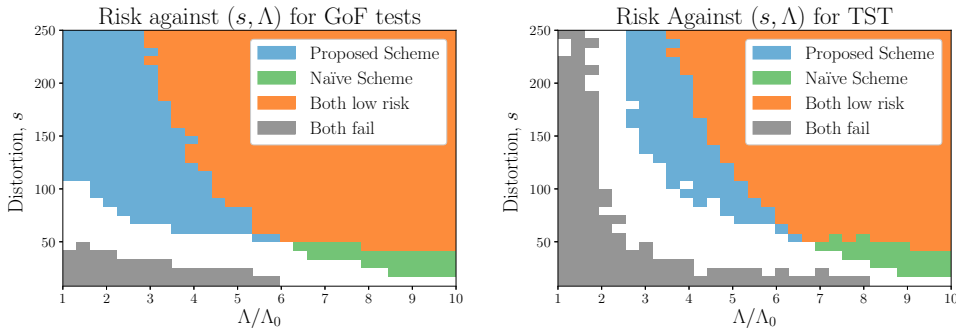

Figure 1: Risks of the proposed tests from sections 2 and 3 for GoF and TST respectively, and the performance of the naïve scheme, on synthetic SBMs with $n = 1000, a/b = 3$. Both schemes attain high risk ($> 1 - \delta$) in the grey region, intermediate risk in the white, and the colours indicate which of the schema attain low risk ($< \delta$), where $\delta = 0.01$ for GoF and $\delta = 0.1$ for TST.

## 4.2 Political Blogs Dataset [AG05]

The political blogs dataset [AG05] is canonical in the study of community detection, and consists of $n = 1222$ nodes. Here, we vary the effective SNR by randomly subsampling the edges of the graphs at rate $\rho$. See Appendix C.2 for further details. In this dataset, the ground truth partition $x_{\text{True}}$ is available, which in turn yields accurate estimates of the connectivity probabilities $(a, b)$. For this graph $a/b \approx 10$. Further, spectral clustering alone incurs $\approx 50$ errors in this graph, which is larger than $\sqrt{1222} \approx 35$. As a consequence, the behaviour in the 'small changes' regime where the test relies on recovery - is not well illustrated in the following.

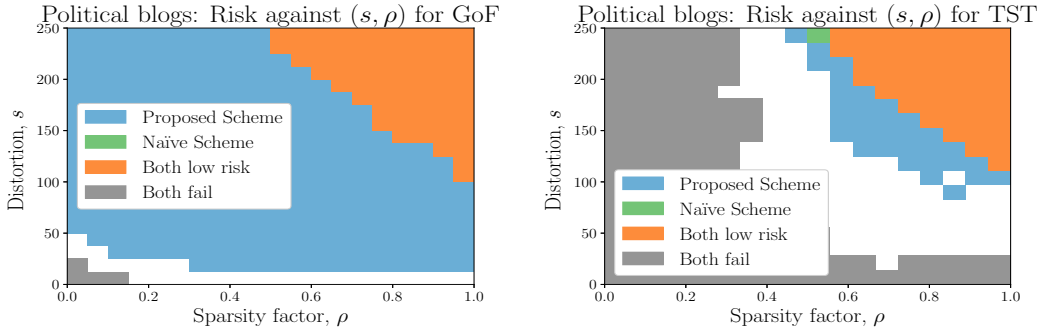

Figure 2: Risks of the tests applied to the Political Blogs graphs - colour scheme is retained from Fig. 1. The X-axis plots the sparsification factor, which serves as a proxy for SNR. Features similar to Fig. 1 can be seen. The GoF plot improves since $a/b$ is bigger, while the TST plot suffers since the political blogs graph is not completely described as a 2-community SBM [Lei16].

**Goodness-of-Fit.** We determine the size of the test by running the GoF procedures against $x_{\text{True}}$. To determine power, we construct a partition $y$ by relabelling a random set of nodes of size $s$, and running the GoF procedures against $y$ *with the same graph*.

**Two-Sample Testing.** We compare the political blogs graph $G$ against two other graphs drawn from SBMs. Size is detemined by drawing $G'$ according to an SBM of community $x_{\text{True}}$ and running the TST procedure, and power is determined by drawing a $y$ as above, generating $H$ according to an SBM of community $y$, and running the TST procedure. Note that this experiment is thus semi-synthetic.

## 4.3 Gaussian Markov Random Fields (GMRFs)

Frequently instead of simply receiving a graph, one receives i.i.d. samples from a graph-structured distribution, and it is of interest to be able to cluster nodes with respect to the latent graph. For example, in large-scale calcium imaging, it is possible to simultaneously record the activity pattern of thousands of neurons, but not their underlying synaptic connectivity [Pne+16]. Here, we explore the behavior of our tests for GMRFs where the underlying graph structure is randomly drawn from an SBM and and we only observe the nodes.

A heuristic reason for why our methods might succeed in such a situation arises from the local tree-like property of sparse random graphs (see, e.g. [DM10]). For graphs with mean degree $d \ll n$, typical nodes do not lie in cycles shorter than $\sim \frac{\log n}{2 \log d}$. In MRFs, this tree-like property induces correlation decay: the correlation between two nodes decays geometrically up to graph-distance $\sim \frac{\log n}{2 \log d}$. Thus, the covariance matrix closely approximates $\sigma_1 G + \sum_{i=2}^{k} (\sigma_1 G)^i + \sigma_0 \mathbf{1}\mathbf{1}^{\mathsf{T}}$ for some $\sigma_0 \ll \sigma_1$, small $k$, and $G$, the adjacency matrix of the graph. Since the local structure of the graph is so expressed, both clustering and testing applied directly to the covariance matrix should be viable.

We report experimentation on the GMRF (see, e.g. [WJ08, Ch. 3]), which comprises random vectors $\zeta \sim \mathcal{N}(0, \Theta^{-1})$, where the non-zero entries of the precision matrix $\Theta$ encode the conditional dependence structure of $\zeta$. Following standard parametrisations [WWR10], we set $\Theta = I + \gamma G$, where $G \sim P(G|x)$ is an adjacency matrix from an SBM with latent parameter $x$, and $\gamma$ is a scalar. Below, we fix the SBM parameters $a, b$ and the level $\gamma$, and explore risks against $s$ and sample size $t$.

Following the above heuristic, we naïvely adapt community recovery and testing to this setting, by replacing all instances of the graph adjacency matrix in previous settings with the sample covariance matrix. Figure 3 presents our simulations of the risk of this test when $n = 1000$, and $(a, b) \approx (12.3 \log n, 1.23 \log n)$, at $\Lambda \approx 9 \log(n)$ (for details see Appx. C.3). This large SNR is chosen so that community recovery would be easy if the graph was recovered;[5] this emphasizes the role of the sample size, $t$. Importantly, in this implementation, the threshold for rejecting the null has been fit using data (unlike in the previous sections). This is since we lack a rigorous theoretical understanding of this problem, and have not analytically derived expressions for the thresholds. As a result, these plots should be treated as speculative research intended to underscore the presence of interesting testing effects in this scenario, and to encourage future work along these lines.

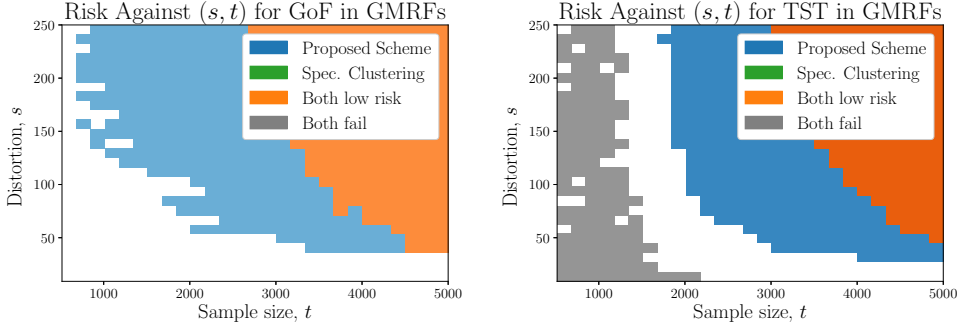

Figure 3: Risks for adaptation of our tests to GMRFs - colour scheme is retained from Fig. 1. The plots show structural similarity to Fig. 1, but with two differences - In GoF, we don't find a high risk region at the sample sizes considered, and the proposed scheme always outperforms the Naïve scheme based on spectral clustering.

## 5   Directions for Future Work

The development of the recovery problem for SBMs suggests a number of directions for further work on the testing problems considered above. For instance, one may investigate the exact constants in the testing threshold that the above work suggests, or one may study the testing problem for SBMs with $k > 2$ communities, which is a practically relevant setting since many real-world networks are significantly better described as $k$-SBMs than as 2-SBMs. In the latter vein, testing problems such as the above may be studied in richer random graph models, such as degree corrected SBMs, or geometric block models. Additionally, testing of strongly imbalanced communities, where one of the communities has size sublinear in $n$ is conceptually unexplored and of interest.

One open problem that draws from the above exposition is if there exists an algorithm for TST in the 2 community setting that does not pass through a partial recovery step and yet works for sparse graphs. We expect that such a method would be necessary for determining exact testing thresholds (for large changes), since the recovery step neccessarily requires some subsampling, which reduces the effective SNR available for testing. In addition, this would be conceptually pleasant, and would eliminate the dissonance in the above work where showing testing guarantees requires passing through recovery guarantees. Such a scheme would also more generally allow study of the testing problem for situations where partial recovery is ill understood.

Finally, we mention that more work is needed on the practical investigation of the effectiveness of the above methods - while the experiments we have run validate the theory, the real-world applicability of the methods above require deeper experimentation. A significant lacuna for this line is the lack of a good real-world dataset for the testing of community changes.

**Acknowledgement**

We thank the anonymous reviewers of this and of previous versions of this paper, particularly one for suggesting the proof of the converse in Theorem 2.

## Footnotes

[1]While our main interest is in the constant degree regime, we also show that testing for small changes is impossible in this setting (e.g Thm 1), and instead logarithmic degrees are needed. Thus, to present our results completely, we must allow $a, b$ to vary at least in the range $[\Omega(1), O(\log(n))]$, or, more succinctly $O(\log n)$. Large scales are not of interest since exact recovery is possible at the logarithmic scale.

[2] Note that $D(G) - G$ is the Laplacian of the graph.

[3] The same also describes the extention of the claims to linearly sized communities

[4]For a proof: since $x, -x$ induce the same law, and since the communities are balanced, for every realization of $G$ such that $\hat{x}$ makes $e_+, e_-$ errors in the community $+, -$ respectively, there is a realization of equal probability where it makes $e_-, e_+$ errors. Further, within community exchangability implies that errors distribute uniformly.

[5]Note, however, we expect graph recovery to be impossible at these sample sizes. Lower bounds from [WWR10] indicate this would require $> 3300$ samples theoretically.

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
