[Supplementary Material]

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

[6] The argument, while simple, gets a little notationally hairy at this point. We recommend that the reader consults Figure 4 frequently, preferably a printed copy that allows one to sketch the various types of connections on it.

[7]This is calculated using a computer algebra system. Analytically it is still easy to argue something similar - for $n > 10$, $\sqrt{n}2^{-0.08n}$ is decreasing. Thus, for $n > 100$, $\sqrt{2n/\pi^2}2^{-0.08n} < {}^{10\sqrt{2}}/_{256\pi} < 1/50$, and thus the expression is at least $e^{9/16} \cdot {}^{49}/_{50} > 1.6 \cdot {}^{49}/_{50} > 1.5$ By following the next footnote with this number, this leads to an analytic proof of the conclusion holding for $\Lambda < {}^7/_{80} \approx 0.087$.

[8]The number 0.074 is calculated using a computer algebra system. Purely analytic calculations are straightforward as well - for example by using $2^{4\tau} \le 1 + 4\tau$ for $\tau < 1/4$, which can be proved by noting that $1 + 4\tau - 2^{4\tau}$ is initially increasing, and then strictly decreasing after a point, and that $1/4$ is a root of this function. This implies that the conclusion holds so long as $\tau < 3/44$, which holds if $\Lambda < 21/176 \approx 0.119$.

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

# Appendix

## A  Proofs omitted in Section 2

### A.1  Proof of Achievability in Theorem 1

We will restrict attention to the case $a > b$ below. The $b > a$ case follows identically. Recall the test in this setting:

$$N_a^{x_0}(G) \underset{H_0}{\overset{H_1}{\gtrless}} \frac{bn}{4} + C_1 \max(\sqrt{nb \log(2/\delta)}, \log(2/\delta)),$$

where $C_1$ is the constant implicit in Lemma 3 below.

Under the null distribution, $N_a^{x_0}(G)$ is distributed as $\mathrm{Bin}(n^2/4, b/n)$, while under the alternate, it is distributed as $\mathrm{Bin}((n-s)^2/4 + s^2/4, b/n) * \mathrm{Bin}(s(n-s)/2, a/n)$. These distributions can be separated by Bernstein concentration bounds [CL06, Ch. 2], as summarised by the following Lemma, which is proved in subsequent sections.

**Lemma 3.** *There exist constants $C_0, C_1 > 1$ such that, if $nb + s(a - b) > C_0 \log(1/\delta)$, then with probability at least $1 - \delta/2$*

*($\alpha$) Under $H_0$: $N_a^{x_0}(G) \le \dfrac{bn}{4} + C_1 \max\left(\sqrt{nb \log(2/\delta)}, \log(2/\delta)\right).$*

*($\beta$) Under $H_1$: $N_a^{x_0}(G) \ge \dfrac{bn}{4} + \dfrac{s(a-b)}{4} - C_1 \sqrt{(nb + s(a-b)) \log(2/\delta)}.$*

As the proof of the above lemma discusses, results of the above type hold in the more generic situation where both the communities and the changes can be unbalanced, so long as each community is of at least linear in $n$ size. This allows one to extend the entirety of this theorem to the setting $n^+ n^- = \Omega(n^2)$ on replacing $bn/4$ above with $\mathbb{E}_{\mathrm{Null}}[N_a^{x_0}(G)]$, where $n^+$ and $n^-$ are the sizes of the two communities, i.e., the number of $i$ such that $x_i = +1$ and $x_i = -1$ respectively.

Since $s|a - b| \ge s\Lambda \ge C \log(2/\delta)$, the lemma above holds in our setting on picking $C$ large enough. ($\alpha$) in Lemma 3 indicates that the false alarm error of test is $\le \delta/2$. Further, since $(nb + s(a - b)) \log(2/\delta) > \log^2(2/\delta)$, part ($\beta$) shows that missed detection error is $\le \delta/2$ if

$$\frac{1}{4}s(a-b) > 2C_1\sqrt{(nb + s(a-b))\log(2/\delta)} \iff \frac{(a-b)^2}{nb + s(a-b)} > C\frac{\log(2/\delta)}{s^2}.$$

The argument is concluded by some casework:

(i) If $nb \le s(a - b)$, then the left hand side of the condition above can be bounded from below by $s(a-b)/2$, and thus $s(a-b) \ge 2C_1 \log(2/\delta)$ is sufficient. But $s(a-b) \ge s(a-b)^2/(a+b) = s\Lambda$ is larger than $C \log(1/\delta)$, and choosing $C$ large enough is sufficient.

(ii) On the other hand, if $nb > s(a - b)$, the left hand side is instead lower bounded by $s^2(a - b)^2/2nb \ge s^2\Lambda/2n$, and thus $s^2\Lambda \gtrsim n \log(2/\delta)$ is sufficient to satisfy the same.  $\square$

### A.1.1  Proof of Lemma 3

The proof proceeds by establishing the centres of the statistic $N_a^{x_0}$ under the null and alternate distributions, and then invoking Bernstein-type bounds [CL06, Ch 2] to show the claimed statements separately.

($\alpha$) For the null, $N_a^{x_0}(G)$ is distributed as $\mathrm{Bin}(n^2/4, b/n)$. Thus, clearly $\mathbb{E}_{\mathrm{Null}}[N] = bn/4$. Further, by Bernstein's inequality for the upper tail,

$$P_{\mathrm{Null}}(N_a^{x_0}(G) > \mathbb{E}_{\mathrm{Null}}[N_a^{x_0}(G)] + nt) \le \exp\left(-\frac{n^2/4 \times t^2/2}{n^2/4 \times (b/n) + nt/3}\right)$$

$$\le \exp\left(-\frac{3}{2}\frac{nt^2}{b + 4t}\right) \le \exp\left(-\frac{3}{8}\frac{nt^2}{t + b}\right).$$

Thus, if

$$\frac{nt^2}{b+t} \ge \frac{8}{3}\log(2/\delta),$$

then this tail has mass at most $\delta/2$. We may now consider the two cases

(i) If $nb \le 16/3 \log(2/\delta)$, then plugging in $t = 16/3 \frac{\log(2/\delta)}{n}$ above yields that the the condition is satisfied, since then

$$\frac{nt^2}{b+t} \ge \frac{nt^2}{2t} = \frac{nt}{2} = \frac{8}{3}\log(2/\delta).$$

(ii) If $nb \ge 16/3 \log(2/\delta)$, then setting $t = \sqrt{(16/3)\frac{b}{n}\log 2/\delta}$ we can bound

$$\frac{nt^2}{b+t} = \frac{16/3 \log(2/\delta)}{1 + \sqrt{(16/3)\log(2/\delta)/nb}} \ge \frac{16/3 \log(2/\delta)}{2}.$$

As a consequence, picking $nt = \max(\sqrt{(16/3)nb\log(2/\delta)}, 16/3 \log(2/\delta))$ implies that the probability in question is at most $\delta/2$.

We note that this calculation can be made more robust, in that if the communities are unbalanced but linearly sized with $n$, then the number of edges crossing is $n^+(n - n^+) = \Omega(n^2)$ in the above, and essentially the same goes through with $n^2/4$ replaced by $n^2/C$ for some constant $C$.

($\beta$) This proof proceeds in much the same way as the above. With the modification that the distribution of $N_a^{x_0}(G)$ is now $\mathrm{Bin}(n^2/4 - s(n-s)/2, b/n) * \mathrm{Bin}(s(n-s)/2, a/n)$, since $2 \times s(n-s)/4$ of the edges are now between nodes of the same communities. The centre of this is easily seen to be $\frac{nb}{4} + \frac{s(n-s)}{2}\frac{a-b}{n}$. Further invoking the Bernstein lower tail, we find that

$$P_{\mathrm{Alt}}(N_a^{x_0}(G) \le \mathbb{E}_{\mathrm{Alt}}[N_a^{x_0}(G)] - nt) \le \exp\left(-\frac{1}{2}\frac{n^2 t^2}{\frac{s(n-s)}{2} \cdot \frac{a}{n} + \frac{n^2 - 2s(n-s)}{4} \cdot \frac{b}{n}}\right)$$

$$\le \exp\left(-\frac{n^2 t^2}{nb + s(a-b)}\right)$$

The required claim now follows directly by setting $t = \sqrt{\frac{(nb + s(a-b))\log(2/\delta)}{n}}$.

Again, the above can also be rendered more robust to imbalance. Suppose that the communities and the changes are both imbalanced, and let $n^+, n^-$ be the sizes of the communities in $x_0$, and $s^+, s^-$ be the number of nodes that are moved from $+$ to $-$ and vice-versa according to the alternate $x$. Then the number of edges which behave according to $a/n$ in the alternate is $\tau = s^+(n^- - s^-) + s^-(n^+ - s^+)$. But $\tau \le s(n^+ + n^- - s^+ - s^-) = sn$, so the concentration results go through with a weakening of a factor of 2. Further, assume wlog that $s^+ \ge s^-$. since $s^+ + s^- = s$, and $n^+ + n^- = n$, we have that

$$\tau = s^+(n-s) + (s - 2s^+)(n^+ - s^+).$$

Minimising the above subject to $s^+ \in [s/2 : s]$, we find that the minima can be uniformly lower bounded by $\min(s\min(n^+, n^-), s(n-s)/2)$. So long as each community is of linear size, this is $\Omega(sn)$, and thus the centre of the statistic moves by $\Omega(s(a-b))$ with respect to the null statistic.

Putting the two effects above together, we can write that under the alternate distribution, with probability $\ge 1 - \delta/2$,

$$N_a^{x_0}(G) \ge \mathbb{E}_{\mathrm{Null}}[N_a^{x_0}(G)] + \frac{1}{C_1}s(a-b) - C_2\sqrt{(nb + s(a-b))\log(2/\delta)}.$$

In conjunction with the discussion for unbalanced but linearly sized communities in case ($\alpha$), the above allows the claims of the achievability part of Theorem 1 to hold for the case where both communities are of linear size and changes are not constrained to be balanced without any change other than a weakening of the constants implicit in the same. The only modification required for this is to update the tests to threshold at $\mathbb{E}_{\mathrm{Null}}[N_a^{x_0}(G)] + (\text{fluctuation term})$ instead of at $bn/4$ as presented in the main text.

## A.2 Proofs of converse bounds from Theorem 1

This section begins with an exposition of Le Cam's method, which is the general proof strategy we employ to show both these converse bounds. This is followed by separate subsections devoted to each converse bound claimed in Theorem 1.

### A.2.1 Le Cam's method.

The generic lower bound strategy is constructed by noting that the minimax risk of the goodness-of-fit problem is lower bounded by the risk of the same with any given prior on the alternate communities, i.e. the risk of the problem

$$H_0 : x = x_0 \quad \text{vs} \quad H_1 : x \sim \pi$$

for a $\pi$ supported on $\{x : d(x, x_0) \geq s\}$ (or some restriction of the same, as in the following sections), and the Bayes risk

$$R_\pi := \inf_{\varphi: G \to \{H_0, H_1\}} P(\varphi = H_1|x_0) + \sum_{x:d(x,x_0)\geq s} \pi(x)P(\varphi = H_0|x).$$

By classical Neyman-Pearson theory [see, e.g., LR06], the likelihood ratio test is optimal under the above risk, and

$$R_\pi = 1 - d_{\text{TV}}\left(P(G|x_0), \langle P(G|x)\rangle_\pi\right),$$

where $\langle P(G|x)\rangle_\pi := \sum_x \pi(x)P(G|x)$, and $d_{\text{TV}}$ is the total variation distance

$$d_{\text{TV}}(P, Q) := \frac{1}{2}\|P - Q\|_1.$$

We proceed by bounding $d_{\text{TV}}$ by an $f$-divergence more conducive to tensorisation in order to exploit the (conditional) independence of the edges in an SBM, and then by choosing an appropriate $\pi$. The $f$-divergence inequalities we use are

1. $\chi^2$ bound: Recall that

$$D_{\chi^2}(Q\|P) = \sum_x P(x)\left(\frac{Q(x) - P(x)}{P(x)}\right)^2 = \mathbb{E}_P[L^2(X)] - 1,$$

   where $L(x) := Q(x)/P(x)$ is the likelihood ratio. It holds that

$$d_{\text{TV}}(P, Q) \leq \sqrt{\frac{1}{2}\log(1 + D_{\chi^2}(Q\|P))},$$

   which follows from Pinsker's inequality and the fact that

$$D_{\text{KL}}(Q\|P) \leq \log(1 + D_{\chi^2}(Q\|P)),$$

   which is a consequence of Jensen's inequality applied to the $\log$ (or, equivalently, the monotonicity of Rényi divergences).
   Invoking the above inequality and Le Cam's method, we find that for any choice of $\pi$, and for $L(G) := \frac{\langle P(G|x)\rangle_\pi}{P(G|x_0)}$, the following is necessary for the minimax risk of the GoF problem to be bounded above by $\delta$ :

$$\mathbb{E}_{x_0}[L^2(G)] \geq \exp\left(2(1 - \delta)^2\right).$$

   For $\delta \leq 1/4$, this yields a necessary lower bound of $\mathbb{E}_{x_0}[L^2] > 3.08$.

2. Hellinger bound: The Bhattacharya coefficient of $P, Q$ is defined as

$$\text{BC}(P, Q) := \sum_x \sqrt{P(x)Q(x)},$$

   and the Hellinger divergence as

$$D_{\text{H}}(P, Q) := \sqrt{1 - \text{BC}(P, Q)} = \frac{1}{\sqrt{2}}\|\sqrt{P} - \sqrt{Q}\|_2.$$

We exploit the relation

$$d_{\mathrm{TV}}(P,Q) \leq \sqrt{D_{\mathrm{H}}^2(P,Q)(2 - D_{\mathrm{H}}^2(P,Q))} = \sqrt{1 - \mathrm{BC}^2(P,Q)},$$

which is a consequence of the Cauchy-Schwarz inequality.

Again plugging this in with $Q = \langle P(G|x) \rangle_\pi$, we find that in order for the risk to be smaller than $\delta$, we must have that

$$\delta \geq 1 - \sqrt{1 - \mathrm{BC}^2} \geq \frac{\mathrm{BC}^2}{2} \implies \mathrm{BC} \leq \sqrt{2\delta},$$

where $\mathrm{BC} = \mathrm{BC}(\langle P(G|x) \rangle_\pi, P(G|x_0))$.

We now proceed to show the claimed bounds. Recall that we are required to show that if $R_{\mathrm{GoF}} < \delta \leq 1/4$, them

$$\Lambda \gtrsim \log(1 + n/s^2) \tag{5}$$
$$s\Lambda \gtrsim \log(1/\delta). \tag{6}$$

### A.2.2 Proof of the converse bound (5)

For convenience, we let

$$\nu := (a - b)^2 \left( \frac{1}{a(1 - a/n)} + \frac{1}{b(1 - b/n)} \right). \tag{7}$$

Since $a, b \leq n/2$, and since $a/b = \Theta(1)$, we have $\Lambda \asymp \nu$, and it suffices to show the same bound on the latter.

We invoke Le Cam's method with a $\chi^2$-bound. Let $m := n/2, t := s/2$ and let $x_0$ be the partition $([1:m], [m+1:2m])$.

The alternate prior is chosen to be the uniform prior on the set of alternate partitions constructed as follows. For each $T \subset [1:m]$, we define the partition

$$y_T(+) = [1:m] \cup (m+T) \sim T$$
$$y_T(-) = [m+1:2m] \cup T \sim (m+T),$$

where $(m+T) = \{i + m : i \in T\}$. Let $\mathcal{Y}_t := \{y_T : T \subset [1:m], |T| = t\}$. For conciseness, we define the measures on $\mathcal{G}$ :

$$P_{y_T}(\cdot) := P_T(\cdot) := P(\cdot \mid y_T),$$

and set $P_0 = P(\cdot \mid x_0)$. Further, for convenience, we set $p = a/n$ and $q = b/n$.

For a graph $G$, we find that $L(G) := \frac{1}{|\mathcal{Y}_t|} \sum_{x \in \mathcal{Y}_t} \frac{P_x(G)}{P_0(G)}$. To invoke Le Cam's method (§A.2.1), we need to upper bound $\mathbb{E}_{P_0}[L^2(G)]$.

To this end, we will define for an edge $e = (u, v)$, and a graph $G$ (which is implicit in the notation)

$$f_e(q, p) := (q/p)^e ((1 - q)/(1 - p))^{1-e}. \tag{8}$$

Above, $f_e(q, p)$ arises as a ratio of the probabilities of a $\mathrm{Bern}(q)$ and a $\mathrm{Bern}(p)$ random variable. Thus, it is the likelihood ratio of an edge being between nodes in the different and in the same community.

First observe that

$$\frac{P_T}{P_0} = \left( \prod_{\substack{i \in [1:m] \sim T, \\ j \in m+T}} f_{ij}(p, q) \right) \left( \prod_{\substack{i \in [m+1:2m] \sim m+T, \\ j \in T}} f_{ij}(p, q) \right) \left( \prod_{\substack{i \in [1:m] \sim T, \\ j \in T}} f_{ij}(q, p) \right) \left( \prod_{\substack{i \in [m+1:2m] \sim m+T, \\ j \in m+T}} f_{ij}(q, p) \right) \tag{9}$$

An important feature of the setup above is that every term in the above product is independently distributed, and wherever $f_{ij}(p, q)$ appears, the corresponding $e_{ij}$ is $\mathrm{Bern}(q)$, and similarly with $f_{ij}(q, p)$ and $\mathrm{Bern}(p)$.

Note that

$$\mathbb{E}_{P_0}[L^2(G)] = \sum_{T_1, T_2 \subseteq [1:m] \text{ of size } t} \mathbb{E}_{P_0}\left[\frac{P_{T_1}(G)P_{T_2}(G)}{P_0^2(G)}\right],$$

and so we must control expectations of this form in order to apply Le Cam's method. Let us fix $T_1$ and $T_2$ for now, and partition the nodes into groups as described by the Figure 4[6].

Figure 4: A schematic of the nodes, partitioned according to their labellings in $x_0, y_{T_1}, y_{T_2}$. The two ovals denote the partition induced by $x_0$ into groups marked $+$ and $-$. The section $1F2F^+$ denotes the nodes in the $+$ group whose labels remain *fixed* to $+$ in both $y_{T_1}, y_{T_2}$. The section marked $1S2F^+$ denotes the nodes in the $+$ group whose labels are *switched* to $-$ in $y_{T_1}$ but remain *fixed* to $+$ in $y_{T_2}$. Other labels are analogously defined.

Note that in the figure, $1F2F^+ = [1:m] \sim (T_1 \cup T_2)$, $1S2S^- = (m+T_1) \cap (m+t_2)$ and so on. Also, importantly, the size of groups with the same number of $S$s and $F$s in the above representation is identical (i.e., $|1F2S^+| = |1F2S^-| = |1S2F^+| = |1S2F^-|$ and so on.)

We consider how the terms relating to the edge $(u,v)$ for any $u,v \in [1:2m]$ appear in the product $\frac{P_{T_1}P_{T_2}}{P_0^2}$. Below,

- Clearly, if $u$ and $v$ are both in the same group in both settings, the behaviour of the edge $(u,v)$ under the alternate distributions and the null distribution is identical, and these terms will not appear in the product.

- If both $(u,v) \in 1F2F^+ \times 1F2F^- \cup 1S2S^+ \times 1S2S^-$, then again, the edge $(u,v)$ has identical distribution under both alternates and the null, and these terms do not appear in the product.

- If $(u,v) \in 1F2F^+ \times 1F2S^+$, then the $(u,v)$ term does not appear in $P_{T_1}/P_0$, but appears once in $P_{T_2}/P_0$. Since likelihoods must average to 1, and since the distributions of the edges are independent, any term which appears just once is averaged out when we take expectations with respect to $P_0$. Thus, even though these terms appear in the product, we may ignore them due to our eventual use of the expectation operator. A quick check will show that the same effect happens for $(u,v) \in \Gamma_1 \times \Gamma_2$, where $\Gamma_1$ can be obtained by inverting one instance of an $F$ to a $S$ or vice versa, and possibly changing the sign (e.g. $1F2S^- \times 1S2S^+$.) Thus, all such pairs can be safely ignored.

- This leaves us with edges of the form $\{1F2F^{\pm} \times 1S2S^{\pm}\} \cup \{1F2S^{\pm} \times 1S2F^{\pm}\}$. In these cases, if the signs of the two choices match - i.e.

$$(u, v) \in \Gamma^{+} \times \widetilde{\Gamma}^{+} \text{ for } (\Gamma, \widetilde{\Gamma}) \in \{(1F2F, 1S2S), (1S2F, 1F2S)\},$$

  then we will obtain a contribution of $f_{uv}(q, p)^2$ to the product. On the other hand, if they differ, then we will obtain a contribution of $f_{uv}(p, q)^2$

Accounting for the above, and taking expectation, we have that

$$\mathbb{E}\left[\frac{P_{T_1} P_{T_2}}{P_0^2}\right] = (\Psi)^{|1F2F^+| \cdot |1S2S^+| + |1F2F^-| \cdot |1S2S^-| + |1S2F^+| \cdot |1F1S^-| + |1F2S^+| \cdot |1S2F^-|}, \quad (10)$$

where

$$\Psi := \mathbb{E}_{e \sim \text{Bern}(p)}[f_e(q, p)^2] \mathbb{E}_{e \sim \text{Bern}(q)}[f_e(p, q)^2] \quad (11)$$

Further, since in our choice of the alternate communities the groups with the same number of $S$s and $F$s have identical size, and thus we may rewrite (10) above as

$$\mathbb{E}\left[\frac{P_{T_1} P_{T_2}}{P_0^2}\right] = \Psi^{2(|1F2F^+||1S2S^+| + |1S2F^+|^2)}.$$

For convenience, let $|1S2S^+| = |T_1 \cap T_2| = k$. We then have that $|1S2F^+| = t - k$ and $|1F2F^+| = m + k - 2t$.

We thus have that

$$\mathbb{E}_{P_0} \frac{P_{T_1} P_{T_2}}{P_0^2} = \exp\left((\log \Psi)(2k(m + k - 2t) + 2(t - k)^2)\right) \quad (12)$$

$$= \exp\left((\log \Psi)(2mk + 2k^2 - 4kt + 2k^2 + 2t^2 - 4kt)\right) \quad (13)$$

$$= \exp\left((\log \Psi)(2mk + 4k^2 - 8kt + 2t^2)\right) \quad (14)$$

$$\leq \exp\left((\log \Psi)((2m - 4t)k + 2t^2)\right), \quad (15)$$

where we have used that $k \leq t$.

Now, for $(p, q) = (a/n, b/n)$,

$$\Psi = \left(\frac{q^2}{p} + \frac{(1 - q)^2}{(1 - p)}\right)\left(\frac{p^2}{q} + \frac{(1 - p)^2}{(1 - q)}\right) \quad (16)$$

$$= \left(1 + \frac{(p - q)^2}{p(1 - p)}\right)\left(1 + \frac{(p - q)^2}{q(1 - q)}\right) \quad (17)$$

$$= \left(1 + \frac{(a - b)^2}{na(1 - a/n)}\right)\left(1 + \frac{(a - b)^2}{nb(1 - b/n)}\right) \quad (18)$$

$$= 1 + \frac{\nu}{n} + O(n^{-2}) \leq 1 + 2\frac{\nu}{n}. \quad (19)$$

As a consequence, using $2m = n$, and the development above,

$$\mathbb{E}_{P_0} \frac{P_{T_1} P_{T_2}}{P_0^2} \leq \exp\left(\frac{4t^2}{n}\nu\right)\exp\left(2k\nu(1 - 4t/n)\right). \quad (20)$$

The above is insular to the precise identities of $T_1, T_2$. Further, for a given $T_1$, the number of partitions $T_2$ such that $|T_1 \cap T_2| = t$ is $\binom{t}{k}\binom{m-t}{t-k}$. Feeding this into the expression for $\mathbb{E}[L^2(G)]$ and some simple manipulations yield that

$$\mathbb{E}_{P_0}[L^2(G)] \leq \frac{e^{\frac{4t^2}{n}\nu}}{\binom{m}{t}} \sum_{k=0}^{t} \binom{t}{k}\binom{m - t}{t - k}\exp\left(2k\nu(1 - 4t/n)\right), \quad (21)$$

where we remind the reader that $t = s/2, m = n/2$.

Recall from §A.2.1 that if $\mathbb{E}_{P_0}[L^2] < 3$, then the risk exceeds 0.25. Thus, we will aim to upper bound (21) by 3.

We begin by rewriting

$$\mathbb{E}_{P_0}[L^2(G)] \leq \frac{e^{\frac{4t^2}{n}\nu}}{\binom{m}{t}} \sum_{k=0}^{t} \binom{t}{k}\binom{m-t}{t-k} \exp\left(2k\nu(1 - 4t/n)\right), \tag{22}$$

$$= e^{\frac{4t^2}{n}\nu}\mathbb{E}[\xi^Z], \tag{23}$$

where $\xi := \exp\left(2\nu(1 - 4t/n)\right) > 1$ and $Z = \sum_{i=1}^{t} Z_i$, where $Z_i$ are sampled without replacement from the collection of $t$ (+1)s and $m - t$ (0)s. Note that $z \mapsto \xi^z$ is continuous and convex for $\xi \geq 1$. By Theorem 4 of [Hoe63],

$$\mathbb{E}[\xi^Z] \leq \mathbb{E}[\xi^{\widetilde{Z}}],$$

for $\widetilde{Z} = \sum_{i=1}^{t} \widetilde{Z}_i$, where $\widetilde{Z}_i$ are drawn by sampling with replacement from the same collection. But $\widetilde{Z}$ is just a Binomial random variable with parameters $(t, t/m)$. Thus, we have that

$$\mathbb{E}_{P_0}[L^2(G)] \leq e^{\frac{2t^2}{m}\nu}\left(1 + \frac{t}{m}\left(\exp\left(2\nu(1 - 2t/m)\right) - 1\right)\right)^t \tag{24}$$

$$\leq \exp\left(2\frac{t^2}{m}\nu + \frac{t^2}{m}\left(\exp\left(2\nu(1 - 2t/m)\right) - 1\right)\right) \tag{25}$$

$$\leq \exp\left(\frac{t^2}{m}\left(2\nu + e^{2\nu} - 1\right)\right) \tag{26}$$

$$\leq \exp\left(2\frac{t^2}{m}\left(e^{2\nu} - 1\right)\right), \tag{27}$$

where the final inequality uses $u < e^u - 1$. Using the above, and noting that $m/2t^2 = n/s^2$, we find that

$$\nu \leq \frac{1}{2}\log\left(1 + \frac{\log(3)n}{s^2}\right) \implies \mathbb{E}_{P_0}[L^2(G)] \leq 3,$$

finishing the argument. □

### A.2.3 Proof of the converse bound (6)

Recall that this part of the theorem claims that if $R_{\text{Gof}} \leq \delta \leq 1/4$, then $s\Lambda \geq C\log(1/\delta)$.

We will again use Le Cam's method (§A.2.1), this time controlling the total variation distance by a Hellinger bound.

Let $x_0 = ([1 : n/2], [n/2 + 1 : n])$ be the null partition, and $\mathcal{Y} := \{y\}$, with $y := ([1 : n/2 - s/2] \cup [n/2 + 1 : n/2 + s/2], [n/2 - s/2 + 1 : n/2] \cup [n/2 + s/2 + 1 : n])$. We let $P_{x_0}(G) := P(G|x_0)$, and similarly $P_y$. Recall from the section on Le Cam's method that the following is a necessary condition for the risk to be smaller than $\delta$

$$\text{BC}(P_{x_0}, P_y) \leq \sqrt{2\delta}.$$

The Bhattacharya Coefficient can be estimated directly in this setting. (We omit the derivation below)

$$\text{BC}(P_y, P_{x_0}) = \left(\sqrt{\frac{ab}{n^2}} + \sqrt{\left(1 - \frac{a}{n}\right)\left(1 - \frac{b}{n}\right)}\right)^{s(n-s)} \tag{28}$$

For $u, v < 3/4$,

$$\sqrt{(1-u)(1-v)} \geq 1 - (u+v)/2 - 2(u-v)^2.$$

Thus

$$\mathrm{BC}(P_y, P_{x_0}) \geq \left(1 - \frac{a+b}{2n} + \frac{ab}{n} - 2\frac{(a-b)^2}{n^2}\right)^{s(n-s)} \qquad (29)$$

$$= \left(1 - \frac{(\sqrt{a} - \sqrt{b})^2}{2n} - 2\frac{(a-b)^2}{n^2}\right)^{s(n-s)} \qquad (30)$$

$$\geq \left(1 - \frac{(\sqrt{a} - \sqrt{b})^2}{n}\right)^{s(n-s)} \qquad (31)$$

$$\geq \exp\left(-2s(\sqrt{a} - \sqrt{b})^2\right), \qquad (32)$$

where the third inequality uses $(a+b) < n/4$, and the final uses used $1 - u \geq e^{-2u}$ for $0 < u \leq 0.75$—which applies since $0 < (\sqrt{a} - \sqrt{b})^2 < \max(a,b) < n/4$—and $n - s \leq n$.

Now note that

$$(\sqrt{a} - \sqrt{b})^2 = \frac{(a-b)^2}{(\sqrt{a} + \sqrt{b})^2} \leq \frac{(a-b)^2}{a+b} = \Lambda,$$

and thus,

$$\mathrm{BC}(P_y, P_{x_0}) \geq \exp\left(-2s\Lambda\right).$$

Invoking the condition for $R_{\mathrm{GoF}} \leq \delta$ above, we have

$$\exp\left(-2s\Lambda\right) \leq \sqrt{2\delta}$$

$$\iff s\Lambda \geq \frac{1}{4} \log \frac{1}{2\delta}.$$

For $\delta \leq 1/4$, we may further lower bound the above by $(\log(1/\delta))/8$. $\qquad\square$

## A.3   A comment on the role of $\Lambda$ when $a/b \neq \Theta(1)$

The main text concentrates on the setting where $a/b$ is a constant. Here, we briefly comment on the setting where the ratio $\rho := \frac{\max(a,b)}{\min(a,b)}$ is *diverging* with $n$. In the setting of balanced communities and divergent $\rho$, the behaviour of the goodness-of-fit problem is no longer described by the quantity $\Lambda = \frac{(a-b)^2}{a+b}$, but instead depends on

$$\mu := \frac{(a-b)^2}{\min(a,b)}.$$

Specifically, our proofs can, with minimal changes, be adapted to say that for balanced GoF, $R_{\mathrm{GoF}}$ can be solved with vanishing risk if the following hold:

$$s\Lambda = \omega(1)$$

$$\mu = \omega(n/s^2),$$

and further, to attain the same, it is necessary to have

$$s\Lambda = \omega(1)$$

$$\mu \gtrsim \log(1 + n/s^2).$$

Indeed, for the lower bounds, $\mu \leq \nu \leq 4\mu$ uniformly, where $\nu$ is the SNR quantity in the previous section, and the upper bounds naturally feature $\mu$.

Together, the above offer a tight characterisation of the GoF problem in the setting of balanced communities and *large s*. Note that $\mu/\Lambda = 1 + \rho$ diverges with $\rho$, and thus the above indicate that GoF testing becomes much easier as this ratio blows up - something to be expected.

Despite the above developments, we concentrated on the setting $\rho = \Theta(1)$ in the main text. This is largely because the majority of the literature on the SBM focuses on this regime, as this is the

hardest setting for inference about the planted structure. Thus, in order to compare to existing work, we examined the $a \asymp b$ setting.

As an aside, we note that unlike the above GoF results, the TST results do not alter in the setting of divergent $\rho$. Theorem 2, and in particular the converse bound $\Lambda \gtrsim 1$, continues to hold for this setting.

On the whole, this line of work is still under investigation, particularly whether the behaviour of GoF for large $\rho$ continues to be driven by $\mu$ in the setting of small changes. We plan to explore this question in later work.

# B  Proofs omitted from section 3

## B.1  Proof of Achievability in Theorem 2

We carry out the analysis for the case $a > b$. The reverse can be handled similarly. Note that this assumption will implicitly be made in all the lemmata that follow.

Recall that the scheme in Algorithm 1 utilises a partial recovery routine. For the purposes of the following argument, we invoke the method of [CRV15], which provides a procedure that, under the conditions of the theorem, that attains with probability at least $1 - 1/n$ recovery with at most $\varepsilon_{\max} n$ errors, where $\varepsilon_{\max} = \min(1/2, 2e^{-C\Lambda})$ for an explicit constant $C$. We choose $\Lambda$ large enough so that $\varepsilon_{\max}$ is bounded strictly below $1/2$ - for convenience, say by $1/3$.

Let $G' \sim P(\cdot|x)$ be an independent copy of $G$, useful in the analysis, and recall the definition of $\widetilde{G}, G_1$ from Algorithm 1. We define the following events that we will condition on in the sequel:

$$\mathcal{E}(G_1) = \{\text{Number of edges in } G_1 \leq an/2\} \qquad \mathcal{E}(\hat{x}) = \{d(\hat{x}, x) \leq \varepsilon_{\max} n\}$$

For succinctness, we let $\mathcal{E} := \mathcal{E}(G_1) \cap \mathcal{E}(\hat{x})$. The analysis proceeds in four steps:

(L1) **Lemma 4.** $P(\mathcal{E}) \geq 1 - 4/3n$.

(L2) **Lemma 5.** $\left| \mathbb{E}[2T^{\hat{x}}(\widetilde{G}) - T^{\hat{x}}(G') \mid \mathcal{E}] \right| \leq a^2$.

(L3) **Lemma 6.** *If* $d(x, y) \geq s$, *then for* $\kappa := (1 - 2\varepsilon_{\max})^2 - 1/(n-1)$,

$$\mathbb{E}[T^{\hat{x}}(G') - T^{\hat{x}}(H) \mid \mathcal{E}] \geq \kappa \frac{(a-b)}{n}(n-s)s.$$

(L4) **Lemma 7.** *Let* $\xi := a^2 + 5\sqrt{2na\log(6n)}$. *Then*

$$P_{\text{Null}}(T \geq \xi | \mathcal{E}) \leq 2/3n$$
$$P_{\text{Alt.}}(T \leq \kappa(a-b)s/2 - \xi | \mathcal{E}) \leq 4/3n$$

We briefly describe the functional roles of the above, and relegate their proofs to the following sections.

(L1) allows us to make use of the typicality of $G_1$ and the recovery guarantees of $\hat{x}$. The former is primarily useful for (L2), while the latter induces (L3).

(L2) lets us avoid the technical issues arising from the fact $\widetilde{G}$ and $G_1, \hat{x}$ are correlated, and allows us to work with the simpler $G'$. It also shows that under the null, the mean of $T$ is small. This lemma is likely loose, and introduces the nuisance condition $a \leq n^{1/3}$.

(L3) shows that under the alternate, the centre of $T$ linearly grows with $s$ despite the weak recovery procedure's errors.

(L4) serves to control the fluctuations in $T$. The $\sqrt{n}$-level term arises from the randomness in $\widetilde{G}, H, G'$, and the $a^2$ term from our use of $G'$ and (L2).

Putting the above together, we find that the risk is bounded by $4/3n + 2/3n + 4/3n \leq 4/n$ if

$$\kappa(a-b)s \geq 4(a^2 + 5\sqrt{2na\log(6n)}).$$

Since $a^2 = a^{3/2}\sqrt{a} \leq \sqrt{na}$, and for $\Lambda$ a large enough constant, $\varepsilon_{\max} \leq 1/3 \implies \kappa \geq (1/3 - 1/(n-1))^2 \geq 1/36$ for $n \geq 7$, the above condition is equivalent to

$$(a-b)s \geq C'\sqrt{na \log(6n)}$$

for a large enough $C'$. Rearranging and squaring, this is equivalent to

$$\frac{(a-b)^2}{a} \gtrsim \frac{n \log(6n)}{s^2}.$$

For $s \geq n^{1/2+c}$ as in the statement, the quantity on the right hand side is decaying with $n$. Further, $\Lambda$ is smaller than the left hand side, so it being bigger than a constant forces the above to hold.

Note that the threshold in Algorithm 1 alters the fluctuation range above from $\sqrt{na}$ to $\sqrt{n(a+b)}$. The reason for this is that this relaxation allows Algorithm 1 to be agnostic to the knowledge of $(a,b)$ - generic spectral clustering schemes do not require this knowledge, and the threshold of our scheme depends only on $n(a+b)$, which can be robustly estimated in our setting since the number of edges in the graph is proportional to this. In addition, invoking the bounds of [CRV15] allows explicit control on $\kappa$ above, and thus provides an explicit value of the constant $C$ in Algorithm 1. $\square$

### B.1.1 Relaxing Exact Balance for TST

We briefly discuss the modifications required to the above analysis in order to extend the same to unbalanced but linearly sized communities. Of the four lemmata used in the proof described above, the proof of Lemma 5 is completely agnostic to the sizes of the communities. In addition, while the proof presented in [CRV15] concentrates on the case of exactly balanced communities, as noted in their Section 1, it can be extended to unbalanced but linearly sized communities with minimal changes, although with a corresponding weakening of the constants in the rate with which error decays with increasing $\Lambda$. This extends Lemma 4 to linearly sized communities.

In contrast, Lemma 6 does rely on the assumption of balance. To sidestep this, we show the following version for use in this setting:

**Lemma 8.** *Let the communities be of sizes $n_+, n_-$. If $d(x,y) \geq s$, then*

$$\mathbb{E}[T^{\hat{x}}(G') - T^{\hat{x}}(H) \mid \mathcal{E}] \geq \frac{(a-b)s}{2}\left(1 - 2\varepsilon\left(\frac{n}{\min(n_+, n_-)} + 2\right)\right).$$

When $\min(n_+, n_-) = cn$, for some $c > 0$, then by enforcing that $\varepsilon$ is smaller than, say, $(3(2 + 1/c))^{-1}$, which may be done by choosing a large enough, but $O(1)$, value of $\Lambda$, the above can be expressed as $\Omega(s(a-b))$.

Lastly, the alternate case in Lemma 7 must be adjusted. However, this concentration result is actually proved by arguing that the event $\{T \leq \mathbb{E}_{\text{Alt}}[T^{\hat{x}}(G') - T^{\hat{x}}(H)|\mathcal{E}] - \xi\}$ has low probability given $\mathcal{E}$, and the above lemma implies that this expectation is at least $\Omega(s(a-b)))$, so the corresponding tail bound goes through to the required form trivially.

At this point, the concluding remarks of the above proof apply to finish the argument.

## B.2 Proofs of Lemmata used in B.1

### B.2.1 Proof of Lemma 4

We first note that by the work of [CRV15], or [FC19], under the conditions of the theorem, $\mathcal{E}(\hat{x})$ holds with probability at least $1 - 1/n$. By a union bound, it suffices to show that $P(\mathcal{E}(G_1)) \geq 1 - \frac{1}{3n}$. Recall that

$$P((e,v) \in G_1|x) = \begin{cases} \frac{a}{2n} & x_u = x_v \\ \frac{b}{2n} & x_u \neq x_v, \end{cases} \tag{33}$$

and that edges are independent. Thus the number of edges in $G_1$ is a sum of Bernoulli random variables of parameter $\leq a/2n$. The factor of 2 arises since $G_1$ is sub-sampled at rate $1/2$. Let $\#G_1$ be the number of edges in $G_1$. We have

$$\mathbb{E}[\#G_1] \leq \binom{n}{2}\frac{a}{2n} \leq \frac{na}{4} \tag{34}$$

$$P(\#G_1 \geq \mathbb{E}[\#G_1] + \sqrt{na \log(3n)}) \leq 1/3n, \tag{35}$$

where the first bound follows from inspection, and the second follows from the Bernstein upper tail bound of [CL06, Ch. 2] and the condition $a \geq 16 \log(6n)/n$. Further invoking this condition we find that $\sqrt{na \log(3n)} \leq na/4$, and thus

$$P(\mathcal{E}(G_1)) = P(\#G_1 \leq na/2) \geq 1 - \frac{1}{3n}. \qquad \square$$

### B.2.2  Proof of Lemma 5

Let

$$c_{uv} := \frac{(a+b) + (a-b)x_u x_v}{2} \leq a.$$

Recall that $c_{uv}/n$ is the probability under $x$ of the edge $(u, v)$ existing.

Also note that for a graph $\Gamma$ and a partition $z$,

$$T^z(\Gamma) = \sum_{1 \leq u < v \leq n} z_u z_v \Gamma_{uv},$$

where $\Gamma_{uv} := \mathbf{1}\{(u, v) \in \Gamma\}$.

We're interested in controlling

$$T = 2T^{\hat{x}}(\widetilde{G}) - T^{\hat{x}}(G') = \sum \hat{x}_u \hat{x}_v (2\widetilde{G}_{uv} - G'_{uv}).$$

Since $\hat{x}$ is a deterministic function of $G_1$, $\widetilde{G}$ is independent of $\hat{x}$ given $G_1$. Further, $G'$ is independent of $(G_1, \widetilde{G})$. Lastly observe that

$$P((u, v) \in \widetilde{G} \mid G_1) = \frac{c_{uv}/2n}{1 - c_{uv}/2n}(1 - (G_1)_{uv}).$$

As a consequence,

$$\mathbb{E}[T \mid G_1] = \sum \hat{x}_u \hat{x}_v \left( 2 \cdot \frac{c_{uv}/2n}{1 - c_{uv}/2n}(1 - (G_1)_{uv}) - \frac{c_{uv}}{n} \right) \tag{36}$$

$$= \sum \hat{x}_u \hat{x}_v \frac{c_{uv}^2/2n^2}{1 - c_{uv}/2n} - \sum \hat{x}_u \hat{x}_v \frac{c_{uv}/n}{1 - c_{uv}/2n}(G_1)_{uv} \tag{37}$$

$$\implies |\mathbb{E}[T \mid G_1]| \leq \sum_{u<v} \frac{c_{uv}^2/2n^2}{1 - c_{uv}/2n} + \sum_{u<v} \frac{c_{uv}/n}{1 - c_{uv}/2n}(G_1)_{uv} \tag{38}$$

$$\leq \frac{a^2/2n^2}{1 - a/2n}\binom{n}{2} + \frac{a/n}{1 - a/2n}\#G_1. \tag{39}$$

where recall that $\#G_1$ is the number of edges in $G_1$. Note that we may condition on $\mathcal{E}$, the occurrence of which is a deterministic function of $G_1$. Since under $\mathcal{E}$ we have $\#G_1 \leq an/2$, we find that

$$|\mathbb{E}[T \mid G_1, \mathcal{E}]| \leq \frac{1}{1 - a/2n}\left( \frac{a^2}{2n^2}\frac{n^2}{2} + \frac{a}{n}\frac{an}{2} \right) \leq a^2, \tag{40}$$

where the final inequality uses that $1/(1 - a/2n) \leq 4/3$, which follows from $a \leq (n/2)^{1/3}$, and $n \geq 2$.

Finally observe that the right hand side of the equation above does not depend on $G_1$. Thus, we may integrate over $P(G_1 \mid \mathcal{E})$ to find that $|\mathbb{E}[T \mid \mathcal{E}]| \leq a^2$.

**Remark**  This lemma is likely rather weak. In particular, the upper bound on $|\mathbb{E}[T|G_1]|$ completely ignores the relationship between $\hat{x}$ & $G_1$, and that between $G_1$ & $c_{uv}$. Indeed, (36) may also be rewritten as

$$\mathbb{E}[T \mid G_1] = \sum \frac{c_{uv}/n}{1 - c_{uv}/2n}\hat{x}_u \hat{x}_v \left( \frac{c_{uv}}{2n} - (G_1)_{uv} \right).$$

Since $(G_1)_{uv} \sim \text{Bern}(c_{uv}/2n)$, and $\hat{x}$ is a clustering derived from $G_1$, it may be possible to control the above to something much smaller than $a^2$. This may require nontrivial use of the $\mathcal{E}(\hat{x})$ conditioning here, which is unused in the above argument. Unfortunately it seems that such control would closely depend on the scheme used to obtain $\hat{x}$, which tend to be complex - most schemes involve non-trivial regularisation of $G_1$, as well as some amount of quantisation of the solution to an optimisation problem to produce $\hat{x}$, due to which the covariance of $G_1$ and $\hat{x}$ is difficult to understand. For completeness' sake we point out that an upper bound on the same of $O(a^2/n)$ would remove the nuisance condition of $a \leq n^{1/3}$ present in Theorem 2. $\qquad \square$

### B.2.3 Setting up the Proof of Lemma 6

We proceed by first developing some intuition behind the proof of Lemma 6 instead of launching straight into the same. Further, we assume throughout that $d(x, y) \geq s$.

Let

$$\text{Incorrect} := \{u \in [1:n] : x(u) \neq \hat{x}(u)\}$$
$$\text{Unchanged} := \{u \in [1:n] : x(u) = y(u)\}.$$

and the sets 'Correct' and 'Changed' be their respective complements. We show in Appendix B.2.4 the following lemma

**Lemma 9.**

$$\mathbb{E}[T^{\hat{x}}(G') - T^{\hat{x}}(H) \mid \hat{x}] = \frac{(a-b)}{n} \Big( n(\text{Unchanged}) - 2n(\text{Incorrect, Unchanged}) \Big)$$
$$\times \Big( n(\text{Changed}) - 2n(\text{Incorrect, Changed}) \Big), \qquad (41)$$

*where*

$$n(\text{Unchanged}) = |\text{Unchanged}|$$
$$n(\text{Incorrect, Unchanged}) = |\text{Incorrect} \cap \text{Unchanged}|,$$

*and the other terms are defined analogously.*

We note that the above lemma holds irrespective of the balance assumptions in the theorem.

Suppose $n(\text{Incorrect}) = k$. Due to the exchangability of the nodes when $|\{u : x(u) = +\}| = |\{u : x(u) = -\}|$, the incorrectly labelled nodes in $\hat{x}$ correspond to a choice of $k \in [0 : n/2]$ nodes picked without replacement from $[1:n]$ uniformly at random. Further, since the changes made in $y$ are chosen independently of the graphs, they are independent of $\hat{x}$. Thus, the number of correct and incorrect nodes changed forms hypergeometric distribution. The expected number of Incorrect nodes changed is precisely $\frac{s}{n} \cdot k$, where $s$ is the number of changes made, and similarly for Incorrect nodes unchanged.

Further invoking the results of [FC19], if $\Lambda \geq C \log(1/\varepsilon_{\max})$, then $k \leq \varepsilon_{max} n$ with probability at least $1 - 1/n$. As a consequence, the bound in Lemma 9 remains large in magnitude even on integrating over the randomness in $\hat{x}$. This was the subject of Lemma 6 from the text, reproduced below for convenience.

**Lemma 6**

$$\mathbb{E}[T^{\hat{x}}(G') - T^{\hat{x}}(H) \mid \mathcal{E}] \geq \left( (1 - 2\varepsilon_{\max})^2 - \frac{1}{n-1} \right) \frac{(a-b)}{n}(n-s)s,$$

the proof of which is the subject of Appendix B.2.5.

### B.2.4 Proof of Lemma 9

We will require explicit counting of a number of groups of nodes. Let us first define them:

Let

$$
\begin{aligned}
S^{++} &:= \{u \in [1:n] : \hat{x}(u) = +1,\ x(u) = +1\}, \quad n^{++} := |S^{++}|,\\
S^{+-} &:= \{u \in [1:n] : \hat{x}(u) = +1,\ x(u) = -1\}, \quad n^{+-} := |S^{+-}|,\\
S^{--} &:= \{u \in [1:n] : \hat{x}(u) = -1,\ x(u) = -1\}, \quad n^{--} := |S^{--}|,\\
S^{-+} &:= \{u \in [1:n] : \hat{x}(u) = -1,\ x(u) = +1\}, \quad n^{-+} := |S^{-+}|.
\end{aligned}
$$

The sets above encode the partitions induced by $\hat{x}$ and $x$, with the first symbol in the superscript denoting the label given by $\hat{x}$. Observe that $S^{+-}, S^{-+}$ are the sets of nodes mislabelled in $\hat{x}$.

Lastly, for $(i,j) \in \{+,-\}^2$, let

$$
\begin{aligned}
C^{i,j} &:= S^{i,j} \cap \{u \in [1:n] : x(u) \neq y(u)\}\\
n_C^{i,j} &:= |C^{i,j}|
\end{aligned}
$$

These are the nodes that change their labels in $y$. Note that the values of each of the above objects is a function of $\hat{x}$. For now we will fix $\hat{x}$, and compute expectations over the randomness in $G', H$ alone.

We first study $N_w$: $N_w^{\hat{x}}(G) = N_w^{\hat{x}}(G[+]) + N_w^{\hat{x}}(G[-])$, where $G[+]$ is the induced subgraph on the nodes $\{u \in [1:n] : \hat{x}(u) = +\}$ and similarly $G[-]$.

By simple counting arguments,

$$
\mathbb{E}[N_w^{\hat{x}}(G'[+]) \mid \hat{x}] = \binom{n^{++} + n^{+-}}{2} \frac{a}{n} - \frac{(a-b)}{n} n^{++} n^{+-}. \tag{42}
$$

Under $H$, the nodes in $C^{++}$ behave as if they were in $S^{+-}$ and those in $C^{+-}$ as if they were in $S^{++}$. Computations analogous to before lead to

$$
\mathbb{E}[N_w^{\hat{x}}(G'[+]) - N_w^{\hat{x}}(H[+]) \mid \hat{x}] = \frac{a-b}{n}\left((n^{++} - n_C^{++}) - (n^{+-} - n_C^{+-})\right)(n_C^{++} - n_C^{+-}) \tag{43}
$$

By symmetry, we can obtain the above for $G[-]$s by toggling the group labels above. Thus, conditioned on a fixed $\hat{x}$, we have

$$
\begin{aligned}
\mathbb{E}[N_w^{\hat{x}}(G') - N_w^{\hat{x}}(H) \mid \hat{x}] = \frac{(a-b)}{n}\Big(&\left((n^{++} - n_C^{++}) - (n^{+-} - n_C^{+-})\right)(n_C^{++} - n_C^{+-})\\
&+ \left((n^{--} - n_C^{--}) - (n^{-+} - n_C^{-+})\right)(n_C^{--} - n_C^{-+})\Big). \tag{44}
\end{aligned}
$$

Similar calculations can be performed for $N_a$. Since in edges across the true partitions, the edges in the same group appear with probability $a/n$ and in different groups with $b/n$, the roles of $a$ and $b$ will be exchanged in this case, leading to a factor of $+(a-b)$ instead of $-(a-b)$. We will suppress the tedious computations, and simply state that

$$
\begin{aligned}
\mathbb{E}[N_a^{\hat{x}}(G') - N_a^{\hat{x}}(H) \mid \hat{x}] = \frac{(a-b)}{n}\Big(&\left((n^{++} - n_C^{++}) - (n^{+-} - n_C^{+-})\right)(n_C^{--} - n_C^{-+})\\
&+ \left((n^{--} - n_C^{--}) - (n^{-+} - n_C^{-+})\right)(n_C^{++} - n_C^{+-})\Big). \tag{45}
\end{aligned}
$$

For convenience, we define

$$
\begin{aligned}
n(\text{Correct, Unchanged}) &:= (n^{++} + n^{--}) - (n_C^{++} + n_C^{--})\\
n(\text{Correct, Changed}) &:= (n_C^{++} + n_C^{--})\\
n(\text{Incorrect, Unchanged}) &:= (n^{+-} + n^{-+}) - (n_C^{+-} + n_C^{-+})\\
n(\text{Incorrect, Changed}) &:= (n_C^{+-} + n_C^{-+})
\end{aligned}
$$

where 'correctness' corresponds to the nodes $u$ such that $\hat{x}(u) = x(u)$, while 'unchangedness' to $u$ such that $x(u) = y(u)$.

Subtracting (45) from (44) then yields that for fixed $\hat{x}$

$$\mathbb{E}[T^{\hat{x}}(G') - T^{\hat{x}}(H) \mid \hat{x}] = \frac{(a-b)}{n}\Big(n(\text{Correct, Unchanged}) - n(\text{Incorrect, Unchanged})\Big)$$
$$\times \Big(n(\text{Correct, Changed}) - n(\text{Incorrect, Changed})\Big). \tag{46}$$

The lemma now follows on observing that

$$n(\text{Unchanged}) = n(\text{Correct, Unchanged}) + n(\text{Incorrect, Unchanged}),$$

and similarly $n(\text{Changed})$. $\hfill\square$

### B.2.5 Proof of Lemma 6

Below we will simply assume that $d(x, y) = s$. The proof is easily extended to $> s$.

Effectively, we are considering the following process: we have a bag of $n$ balls - corresponding to the nodes - of two colours (types), Changed and Unchanged, and we are picking $k \leq n/2$ of them uniformly at random without replacement. Let

$$\eta_1 := n(\text{Unchanged, Incorrect}) \tag{47}$$
$$\eta_2 := n(\text{Changed, Incorrect}) \tag{48}$$

and

$$\zeta := (n(\text{Unchanged}) - 2n(\text{Incorrect, Unchanged}))(n(\text{Changed}) - 2n(\text{Incorrect, Changed}))$$
$$= (n - s - 2\eta_1)(s - 2\eta_2). \tag{49}$$

We now condition on the number of errors being $k$, which imposes the condition that $\eta_1 + \eta_2 = k$. Recall the sampling without replacement distribution, which implies that

$$P(\eta_1 = k - j, \eta_2 = j \mid d(\hat{x}, x) = k) = \frac{\binom{n-s}{k-j}\binom{s}{j}}{\binom{n}{k}}. \tag{50}$$

Thus,

$$\mathbb{E}[\eta_1 | d(x, \hat{x}) = k] = \frac{k}{n}(n - s)$$
$$\mathbb{E}[\eta_2 | d(x, \hat{x}) = k] = \frac{k}{n}(s)$$
$$\mathbb{E}[\eta_1\eta_2 | d(x, \hat{x}) = k] = (n - s)(s)\frac{k(k-1)}{n(n-1)} = s(n - s)\left(\frac{k^2}{n^2} - \frac{k(n-k)}{n^2(n-1)}\right).$$

As a consequence, we obtain that

$$\mathbb{E}[\zeta | d(x, \hat{x}) = k] = s(n - s)\left(1 - 4\frac{k}{n} + 4\frac{k^2}{n^2} - 4\frac{k(n-k)}{n^2(n-1)}\right)$$
$$= s(n - s)\left(\left(1 - 2\frac{k}{n}\right)^2 - 4\frac{k(n-k)}{n^2(n-1)}\right) \tag{51}$$

Note that the above is decreasing as $k$ increases for $k \leq n/2$.

Note further that the Markov chain $\zeta$–$d(\hat{x}, x)$–$G_1$ holds. Thus the above also holds for $\mathbb{E}[\zeta \mid \mathcal{E}(G_1), d(x, \hat{x}) = k]$.

We now condition on $\mathcal{E}(\hat{x})$ to find that

$$\frac{\mathbb{E}[\zeta \mid \mathcal{E}(\hat{x}), \mathcal{E}(G_1)]}{s(n-s)} \geq \left( (1-2\varepsilon_{\max})^2 - 4\frac{\varepsilon_{\max}(1-\varepsilon_{\max})}{n-1} \right) \tag{52}$$

$$\geq (1-2\varepsilon_{\max})^2 - \frac{1}{n-1} \tag{53}$$

where we have used $\varepsilon_{\max} \leq 1/2$, and the (unstated but obvious) condition that $n \geq 2$.

Applying the above to the result of Lemma 9, we find that

$$\mathbb{E}[T^{\hat{x}}(G') - T^{\hat{x}}(H) \mid \mathcal{E}] \geq \left( (1-2\varepsilon_{\max})^2 - \frac{1}{n-1} \right) \frac{(a-b)}{n}(n-s)s. \qquad \square$$

### B.2.6  Proof of Lemma 8

For continuity of exposition, we prove Lemma 8 before Lemma 7.

Below we will simply assume that $d(x,y) = s$. The proof is easily extended to the same being $> s$.

We begin with recalling Lemma 9, and noting that it's proof does not utilise the exact balance assumption. We begin as in the proof of Lemma 6, by defining

$$\eta_1 := n(\text{Unchanged, Incorrect})$$
$$\eta_2 := n(\text{Changed, Incorrect})$$

and noting from Lemma 9 that $\mathbb{E}[T^{\hat{x}}(G') - T^{\hat{x}}(H) \mid \hat{x}] \geq \frac{(a-b)}{n}(n-s-2\eta_1)(s-2\eta_2) =: \frac{(a-b)}{n}\zeta$. Once again, let's fix the errors number of errors made by $\hat{x}$ as some $k$, and let $s_+, s_-$ be the number of changes made in communities $+$ and $-$ respectively. Note that $s_+ + s_- = s$.

Using the above definitions,

$$\zeta = (n-s)(s-2\eta_2) - 2s\eta_1 + 4\eta_1\eta_2 \geq \frac{n}{2}(s-2\eta_2) - 2ks,$$

by noting that $\eta_1\eta_2 \geq 0, \eta_1 \leq k$ and that $s \leq n/2$. Thus,

$$\mathbb{E}[\zeta|d(\hat{x},x) \leq k] \geq \frac{ns}{2}\left( 1 - \frac{4k}{n} - 2\frac{\mathbb{E}[\eta_2|d(\hat{x},x) \leq k]}{s} \right).$$

Suppose that the recovery procedure makes $(k^+, k^-)$-errors in communities $+$ and $-$ respectively, with $k^+ + k^- \leq k$. *Within-community* exchangability of nodes implies that the errors made within a community must be uniformly distributed over the community. Since the changes are made independently of these errors, we must have that the number of changed nodes in community $* \in \{+1, -1\}$ that are incorrectly inferred must be $\mathrm{Hyp}(n_*, s_*, k^*)$ distributed. In particular, this yields that

$$\mathbb{E}[\eta_2|(k^+, k^-)] = \frac{s_+}{n_+}k^+ + \frac{s_-}{n_-}k^- \leq \frac{s_+k^+ + s_-k^-}{\min(n_+, n_-)} \leq \frac{k^+ + k^-}{\min(n_+, n_-)}s.$$

The above immediately yields that

$$\mathbb{E}[\zeta|d(\hat{x},x) \leq k] \geq \frac{ns}{2}\left( 1 - \frac{4k}{n} - 2\frac{k}{\min(n_+, n_-)} \right).$$

On $\mathcal{E}, k \leq \varepsilon n$, leading to the claimed bound. $\qquad \square$

### B.2.7  Proof of Lemma 7

Recall the notation from Appendix B.2.2. Under the null $H \overset{\text{law}}{=} G'$. Below, we will use $G'$ as a proxy for $H$ in the null distribution, and use $H$ only in the alternate.

To begin with, observe that both $G', H$ are independent of $G_1, \widetilde{G}, \hat{x}$, and that $\widetilde{G}$ is independent of $\hat{x}$ given $G_1$. Now, $T^{\hat{x}}$ is a signed sum of independent Bernoulli random variables with parameters

smaller than $a/n$ given $G_1$.. Thus, invoking results from Ch. 2 of [CL06] (and using that for $a \geq C$ for some large enough $C$ implies that $a \geq 16 \log(6n)/n \iff 1/6n \leq \exp(-na/16))$), we find that for $\Gamma \in \{\widetilde{G}, G', H\}$,

$$P\left(\left|T^{\hat{x}}(\Gamma) - \mathbb{E}[T^{\hat{x}}(\Gamma) \mid G_1, \mathcal{E}]\right| \geq \sqrt{2na \log(6n)} \mid G_1, \mathcal{E}\right) \leq \frac{1}{3n},$$

where we have used that $\mathcal{E}$ is determined given $G_1$ (i.e. $\mathcal{E}$ lies in the sigma-algebra generated by $G_1$.)

We now control the null and alternate fluctuations given $\mathcal{E}$.

Null: By the union bound, we find that

$$P\left(2T^{\hat{x}}(\widetilde{G}) - T^{\hat{x}}(G') \geq \mathbb{E}[2T^{\hat{x}}(\widetilde{G}) - T^{\hat{x}}(G') \mid G_1, \mathcal{E}] + 3\sqrt{2na \log(6n)} \mid G_1, \mathcal{E}\right) \leq \frac{2}{3n}$$

Recall from equation (40) from the proof of Lemma 5 that $\mathbb{E}[2T^{\hat{x}}(\widetilde{G}) - T^{\hat{x}}(G') \mid G_1, \mathcal{E}] \leq a^2$. Feeding this in, we find that

$$P\left(2T^{\hat{x}}(\widetilde{G}) - T^{\hat{x}}(G') \geq a^2 + 3\sqrt{2na \log(6n)} \mid G_1, \mathcal{E}\right) \leq \frac{2}{3n}.$$

The right hand side above does not depend on $G_1$, and neither does the fluctuation radius wihtin the probability. Thus integrating over $P(G_1 \mid \mathcal{E})$, we find that

$$P\left(T \geq a^2 + 3\sqrt{2na \log(6n)} \mid \mathcal{E}\right) \leq \frac{2}{3n},$$

where we have used that $T = 2T^{\hat{x}}(\widetilde{G}) - T^{\hat{x}}(H) \overset{\text{law}}{=} 2T^{\hat{x}}(\widetilde{G}) - T^{\hat{x}}(G')$ under the null.

Alt: Following the above development again, this time with lower tails, we find that given $G_1$ with probability at least $1 - 2/3n$,

$$2T^{\hat{x}}(\widetilde{G}) - T^{\hat{x}}(G') \geq -(\mathbb{E}[2T^{\hat{x}}(\widetilde{G}) - T^{\hat{x}}(G') \mid G_1, \mathcal{E}]) - 3\sqrt{2na \log(6n)}$$
$$T^{\hat{x}}(G') - T^{\hat{x}}(H) \geq +(\mathbb{E}[T^{\hat{x}}(G') - T^{\hat{x}}(H) \mid G_1, \mathcal{E}]) - 2\sqrt{2na \log(6n)}$$

Further, given $(G_1, \mathcal{E})$, by Lemmas 5, 6 we have

$$2T^{\hat{x}}(\widetilde{G}) - T^{\hat{x}}(G') \geq -a^2 - 3\sqrt{2na \log(6n)}$$
$$T^{\hat{x}}(G') - T^{\hat{x}}(H) \geq +\kappa(a - b)s(1 - s/n) - 2\sqrt{2na \log(6n)},$$

where $\kappa = (1 - 2\varepsilon_{\max})^2 - 1/(n-1)$. Adding the above, we find by the union bound that

$$P\left(2T^{\hat{x}}(\widetilde{G}) - T^{\hat{x}}(H) \geq \kappa(a - b)s(1 - s/n) - a^2 - 5\sqrt{2na \log(6n)} \mid G_1, \mathcal{E}\right) \geq 1 - \frac{4}{3n}.$$

The claim follows on noting that the right hand side and the fluctuation radius do not depend on $G_1$, and integrating the inequality over $G_1$. $\qquad \square$

## B.3 Proof of the converse bound from Theorem 2.

We restate the lower bound below as a proposition:

**Proposition** *There exists a universal constant $C$, and another $c < 1$ that depends on $C$, such that if $\Lambda \leq C$ and $s \leq \frac{n}{2}(1-c)$, then reliable two-sample testing of balanced communities for $s$ changes is impossible for large enough $n$.*

*In particular, for $a + b < n/4$, the statement holds with $C = 1/8, c = 1/6$ for $n \geq 136$, and in this case, $R_{\text{TST}} \geq 0.25$.*

*Proof.* The proof proceeds by using a variation of Le Cam's method, and importing impossibility results for the so-called distinguishability problem [Ban+16]. In particular, suppose that in the null distribution, the communities are drawn according to the uniform prior on balanced communities, denoted by $\pi$. Further, assume that if a $s$-change is made, then the resulting community is chosen uniformly from all communities that are at least $s$ far from the null community. We have the hypothesis test:

$$H_0 : (G, H) \sim \sum_{x \in \mathcal{B}} \pi_x P(G|x) P(H|x) \quad \text{vs } H_1 : (G, H) \sim \sum_{x,y \in \mathcal{B}} \pi_x \pi_{y|x} P(G|x) P(H|y),$$

where we use $\mathcal{B}$ to denote the set of balanced communities, and $\pi_{y|x}$ is the uniform distribution on $\mathcal{B} \cap \{y : d(x,y) \geq s\}$. For succinctness, let us denote the null and alternate distributions above as $p_{\text{null}}$ and $p_{\text{alt}}$ respectively.

Once again, by Neyman-Pearson theory,

$$R_{\text{TST}} \geq R_\pi \geq 1 - d_{\text{TV}}(p_{\text{null}}, p_{\text{alt}}) \geq 1 - d_{\text{TV}}(p_{\text{null}}, Q) - d_{\text{TV}}(Q, p_{\text{alt}}),$$

where $Q$ is any distribution, and the last inequality is since $d_{\text{TV}}$ is metric.

We choose $Q$ to be the unstructured distribution induced by an Erdős-Rényi graph of parameter $(a+b)/2n$. The primary reason for this is that explicit control on the total variation distance between $p_{\text{null}}$ and $Q$ is then available - for instance, by [WX18, §3.1.2], we have

$$D_{\chi^2}(p_{\text{null}} \| Q) + 1 \leq \mathbb{E}\left[\exp\left(\tau\left(\frac{4\mathscr{H} - n}{\sqrt{n}}\right)^2\right)\right],$$

where $\mathscr{H}$ is a Hypergeometric$(n, n/2, n/2)$ random variable, and

$$\tau = \frac{(a-b)^2}{2(a+b)} + \frac{(a-b)^2}{2(2 - a/n - b/n)}.$$

Notice the extra factor of 2 compared to the expressions in [WX18], which arises since we sum over two independent graphs $G, H$ and not one. We observe that

$$\tau = \Lambda \frac{n}{2n - a - b},$$

and explicitly, if $a + b \leq n/4$, then $\tau \leq \frac{4}{7}\Lambda$.

We now consider the alternate term. As a preliminary, let

$$\gamma := \frac{\sum_{k=0}^{s-1} \binom{n/2}{k/2}^2}{\binom{n}{n/2}}.$$

Note that $\gamma$ is the probability that two balanced communities chosen independently and uniformly, lie within distortion $s$. Indeed, since communities are formed by identifying antipodal points in the boolean cube, the probability of picking a community at distortion $< s$ coincides with that of picking a balanced vector at Hamming distance $< s$ from a given balanced vector in the cube $\{0,1\}^n$. The denominator in $\gamma$ is clearly the number of balanced vectors in the cube, while the numerator is the number of balanced vectors at a distance of $< s$ from any given balanced vector - we choose $k < s$, and choose $k/2$ points marked 1 and $k/2$ marked 0, and flip them all.

As a consequence, we find that for any $x, y \in \mathcal{B}$,

$$\pi_{y|x} \leq \frac{\pi_y}{1 - \gamma}.$$

Thus, in the $\chi^2$ expressions for $p_{\text{alt}}$, we have

$$\mathbb{E}_{(G,H) \sim Q^{\otimes 2}}[(p_{\text{alt}}/Q)^2] = \sum_{x,y,x',y'} \mathbb{E}\left[\frac{P(G|x)P(G|x')}{Q^2(G)} \frac{P(H|y)P(H|y')}{Q^2(H)}\right] \pi_x \pi_{x'} \pi_{y|x} \pi_{y'|x'}$$

$$\leq \frac{1}{(1-\gamma)^2} \sum_{x,y,x',y'} \mathbb{E}\left[\frac{P(G|x)P(G|x')}{Q^2(G)} \frac{P(H|y)P(H|y')}{Q^2(H)}\right] \pi_x \pi_{x'} \pi_y \pi_{y'}$$

$$= \frac{1}{(1-\gamma)^2}\left(1 + \chi^2(\sum_{x \in \mathcal{B}} \pi_x P(G|x) \| Q(G))\right)^2$$

Since the final quantity is explicitly controlled in the cited section, we also have

$$1+D_{\chi^2}(p_{\text{alt}}\|Q) \leq \frac{1}{(1-\gamma)^2}\mathbb{E}\left[\exp\left(\frac{\tau}{2}\left(\frac{4\mathscr{H}-n}{\sqrt{n}}\right)^2\right)\right]^2 \leq \frac{1}{(1-\gamma)^2}\mathbb{E}\left[\exp\left(\tau\left(\frac{4\mathscr{H}-n}{\sqrt{n}}\right)^2\right)\right],$$

the final relation arising from Jensen's inequality.

Since the quantity appears often, we let

$$\beta := \mathbb{E}\left[\exp\left(\tau\left(\frac{4\mathscr{H}-n}{\sqrt{n}}\right)^2\right)\right].$$

Invoking the inequality $d_{\text{TV}} \leq \sqrt{\log(1+D_{\chi^2})/2}$, we find that

$$R_{\text{TST}} \geq 1 - \sqrt{\log(\beta)/2} - \sqrt{\log(\beta(1-\gamma)^{-2})/2} = 1 - \sqrt{\log(\beta/(1-\gamma))}.$$

Note that the only $s$-dependent term in the above bounds is $\gamma$. We first offer control on the $\gamma$, and claim that for $s/n < 1/2$, $\gamma \to 0$. Indeed, since $s \leq n/2$, and by standard refinements of Stirling's approximation (for instance, we use [Gal68, Exercise 5.8] below),

$$\gamma \leq s\frac{\binom{n/2}{s/2}^2}{\binom{n}{n/2}} \leq s\frac{1}{2\pi}\frac{n/2}{s/2(n-s)/2}2^{nh_2(s/n)}\left(\sqrt{\frac{n}{8(n/2)^2}}2^n\right)^{-1}$$

$$\leq \sqrt{\frac{2n}{\pi^2}}2^{-n(1-h_2(s/n))},$$

where $h_2$ is binary entropy in bits.

At this point the argument in the limit as $n \to \infty$ is complete - since $4(\mathscr{H}-n)/\sqrt{2n} \overset{\text{Law}}{\to} \mathcal{N}(0,1)$, $\beta$ is bounded as $n \to \infty$ by $\sqrt{1-2\tau}$ if $\tau < 1/2$, and since in this limit $\tau \to \Lambda/2$ (for $a,b = o(n)$), we obtain that if $\limsup s/n < 1/2$, and $\Lambda < 1$, then $\liminf R_{\text{TST}} > 0$.

Non-asymptotic bounds can be recovered by giving up space on the constants, leading to the statement we have claimed.

Concretely, to attain $R_{\text{TST}} > 1/4$, it suffice to show that $\beta(1-\gamma)^{-1} < e^{9/16}$. Now, for $s < n/3$, we have

$$(1-\gamma)e^{9/16} \geq \left(1 - \sqrt{2n/\pi^2}2^{-0.08n}\right)e^{9/16} > 1.75$$

for $n \geq 136$.[7] Thus, it suffices to control $\beta$ to below 1.75 in this regime. To this end, note that $u \mapsto \exp\left(\tau((4u-n)/\sqrt{n})^2\right)$ is a continuous, convex map, and thus, by [Hoe63, Thm. 4],

$$\beta \leq \mathbb{E}\left[\exp\left(\tau\left(\frac{4\mathscr{B}-n}{\sqrt{n}}\right)^2\right)\right],$$

where $\mathscr{B} \sim \text{Bin}(n/2, 1/2)$.

By Chernoff's bound, $P(|\mathscr{B}-n/4| \geq \sqrt{n}u) \leq 2e^{-4u^2}$, and thus, we have

$$\beta \leq \int_0^\infty P\left(\exp\left(\tau\left(\frac{4\mathscr{B}-n}{\sqrt{n}}\right)^2\right) \geq u\right) du$$

$$\leq \int_0^\infty \min(1, 2u^{-1/4\tau}) d\tau$$

$$= \frac{2^{4\tau}}{1-4\tau},$$

the final equality holding so long as $1/4\tau > 1 \iff \tau < 1/4$. The original claim follows if

$$\frac{2^{4\tau}}{1 - 4\tau} \le \frac{7}{4},$$

which is true for $\tau < 0.074$. Since $\tau \le {^4/_7}\Lambda, \Lambda < 1/8$ implies that $\tau < 4/56 < 0.072$.[8]     □

A couple of quick comments are useful here:

1. Note that the above cannot be applied usefully to GoF. This is because in GoF, the null is explicitly available, and we do not have the benefit of averaging with $\pi$ in the TV expressions. This causes the equivalent term $\chi^2(P(G|x_0)\|Q(G))$ to grow exponentially with $n\Lambda$.

2. The above characterises the tightness of our claimed bounds for TST of large changes - the method works if $\Lambda = \Omega(1)$ and $s \gg \sqrt{n \log n}$, and by the above argument, no test can work if $\Lambda \ll 1$, as long as the change is not extreme ($\limsup s/n < 1/2$).

3. While the above approach is wasteful in how it utilises $s$, this is actually a non-issue, since the bounds require a separate control on $d_{\mathrm{TV}}(p_{\mathrm{null}}\|Q)$, which can only be controlled if $\Lambda = O(1)$. In particular, we cannot pull out better bounds for the small $s$ situation from the above.

## C    Experimental Details

### C.1    Experiments on SBMs

The experiemnts simulate an ensemble of GoF and TST test and evaluate the performance of the two schemes using the sum of false alarm and missed detection probabilities ($FA + MD$).

While the GoF scheme is implemented precisely as in the main text, the experiments use a slightly modified version of Algorithm 1 for the TST:

(i) $G_1$ subsamples $G$ at a rate $\eta$, and the test statistic $T$ is appropriately modified: $T :=$ $\frac{1}{1-\eta}T^{\hat{x}_1}(\widetilde{G}) - T^{\hat{x}_1}(H)$. Intrinsically, the spectral clustering step is the more singal-sensitive part of the scheme 1. While splitting the graphs equally is fine for theoretical results, it is better in practice to devote more SNR to the clustering step, and less to compute the test statistic, which can be done by increasing $\eta$. In the following, we set $\eta = 0.85$. Other values of $\eta$ are explored in Appendix C.1.2.

(ii) The constant factor in the threshold developed in the test is conservative, and we vary it to adjust for different values of $\eta$ and to mitigate its suboptimality. In the experiments, we used the threshold $\frac{3}{4}\sqrt{n(a+b)\log(6n)}$.

As noted in the main text, the experiments are performed for various $(s, \Lambda)$ for a fixed value of $a/b = 3$. $\Lambda$ is varied between $\Lambda_0$ and $10\Lambda_0$ for $\Lambda_0 = 3/4 log(n/100) \approx 1.7$. This is significantly below the theoretical threshold of 2 necessary for non-trivial recovery. Further, $8\Lambda_0 = 2\log(n)$, at which point recovery with constant order distortion becomes viable.

#### C.1.1    Implementation details

The experiment is setup as follows:

1. We fix a value of $\Lambda_0 = 3/4\log(n/100)$ as above. Then, for some choice of $b/a = r$, we choose $(a, b)$ satisfying $\Lambda = \alpha\Lambda_0$ and $\alpha \in [1, 10]$. $r$ is set to be $1/3$.

2. For a fixed number of nodes, $n$, and for $s \in [1 : n/2]$, we consider the balanced partition $x = [x_i]_{i=1}^n$ with

$$x_i = \begin{cases} 0, & 0 \le i \le n/2 \\ 1, & n/2 < i \le n \end{cases}$$

for the null distribution, and the shifted balanced partition $y = [y_i]_{i=1}^n$ with

$$y_i = \begin{cases} 0, & s/2 < i \le n/2 + s/2 \\ 1, & i \in (n/2, n] \cup [0, s/2] \end{cases}$$

for the alternate distribution. This ensures that $d(x, y) = s$. We take $\lfloor \cdot \rfloor$ whenever $s$ or $n$ are odd.

3. We sample $G, G' \sim P(\cdot \,|\, x)$ and $H \sim P(\cdot \,|\, y)$, where $P$ represents drawing from an SBM with parameters $n$, $a$ and $b$, as described in §1.

**GoF procedure.** Recall that we are given a proposed partition $x_0$. Here we set $x_0 = x$. The results of running the tests on the graph $G$ then serve to characterise size, and those on $H$ serve to characterise power.

1. For the naïve scheme, we produce partitions $\hat{x}$ and $\hat{y}$ from $G$ and $H$ respectively via spectral clustering (see below for details), and declare for null in either case if $d(x_0, z) < s/2$, where $z$ is respectively $\hat{x}$ and $\hat{y}$.

2. For the alternate scheme, we instead compute the statistic from §2, and reject on the basis of the threshold developed there.

**TST procedure.** Similarly to the above, runs on the pair $(G, G')$ serve to characterise size, and on $(G, H)$ serve to characterise power of the test. Precisely:

1. For the naïve two-sample test based on recovery and comparison, we estimate $\hat{x}, \hat{x}'$ and $\hat{y}$ from $G$, $G'$ and $H$ respectively. The structure is estimated using spectral clustering (see below for implementation details). We declare that a change has occurred if $d(\hat{x}, \hat{x}') \ge s/2$, and no change if $d(\hat{x}, \hat{y}) < s/2$. We get a false alarm every time we declare a change on the pair $(G, G')$, and we miss a detection whenever we declare no change on the pair $(G, H)$. The false alarm and missed detection probabilities are estimated as an average over $M = 100$ samples.

2. For the two-sample test based on Algorithm 1, we follow the algorithm as stated, making only the modifications previously described. To be precise, we estimate $\hat{x}_1$ from $G_1$, a subsampling of (the edges of) $G$ at rate $\eta$. Then, we compute the test statistics in the null and alternate distributions:

$$T_{\text{Null}} = \frac{1}{1 - \eta} T^{\hat{x}_1}(\widetilde{G}) - T^{\hat{x}_1}(G')$$

and

$$T_{\text{Alt.}} = \frac{1}{1 - \eta} T^{\hat{x}_1}(\widetilde{G}) - T^{\hat{x}_1}(H),$$

where $\tilde{G} = G - G_1$.

In both the above cases, the simulations are performed over a range of $\Lambda = \alpha \Lambda_0$ and $s$, where $\alpha \in [1, 10]$ and $s \in (0, 250)$. Performance is indicated using the sum of false alarm and missed detection rates.

Details associated with the implementation of the aforementioned schemes are given below:

1. All experiments were implemented in the Python programming language (v3.5+), using the Numpy (v1.12+) and Scipy (v0.18+) scientific computing packages [Oli06; JOP01].

2. Structure learning was performed using the Spectral Clustering [Lux07] algorithm, as implemented by the Scikit-learn package (v0.19.1+) [Ped+11].

3. Spectral Clustering was regularized in the manner suggested by [JY16]. Effectively, if $G$ was the adjacency matrix to be submitted to the Scikit-learn spectral clustering function, we performed pre-addition, and instead passed $G + \tau \mathbf{1}\mathbf{1}^\mathsf{T}$. We set $\tau = \frac{1}{10n}$, which proved sufficient to run the spectral clustering function with no errors or warnings.

4. All plots were generated using Matplotlib (v2.1+) [Hun07].

(a) Naïve two-sample test based on structure learning

(b) Two-sample test based on Algorithm 1 for $\eta = 0.7$

(c) Two-sample test based on Algorithm 1 for $\eta = 0.8$

(d) Two-sample test based on Algorithm 1 for $\eta = 0.9$

Figure 5: A comparison between the naïve two-sample test based on structure learning, and the two-sample test we propose in Algorithm 1, for $\eta \in \{0.7, 0.8, 0.9\}$. Error rates lower than $\delta = 0.1$ have been shaded blue to represent "success", while those higher than $1 - \delta = 0.9$ have been shaded orange to represent "failure".

### C.1.2 Modifications to $\eta$

For completeness, we demonstrate how the performance of the modified two-sample test based on Algorithm 1 varies as $\eta$ is changed. Figure 5 compares the naïve two-sample test against the scheme based on Algorithm 1, for three different values of $\eta$: 0.7, 0.8 and 0.9.

We use the following parameters: $n = 500$, $\text{SNR}_0 = \frac{3}{8} \log(n/100) = \frac{3}{8} \log 5 \approx 0.5$, $\frac{b}{a} = r = 1/3$. For $\eta = 0.7$ and $\eta = 0.8$, the threshold used is $\sqrt{n(a + b) \log(6n)}$, while for $\eta = 0.9$, we used a higher threshold of $\frac{3}{2}\sqrt{n(a + b) \log(6n)}$.

While differences are rather subtle, a careful examination may reveal that as $\eta$ increases, the failure region recedes, while the success region advances in the high-$s$, low-SNR regime. However, the cost of this is an increased threshold to maintain success at $\delta = 0.1$, and a wider transition region, indicating that different $\eta$ might be optimal at different $n$.

### C.2 Experiments on the Political Blogs dataset

While the original graph has 1490 nodes, we followed standard practice in selecting the largest (weakly) connected component of the graph, which contains 1222 nodes. We denote this graph as $G$. The true partition of the blogs according to political leaning is available, denoted $x_{\text{True}}$ here. This also allows accurate estimates of the graph parameters $(a, b)$ to be made, and we use these estimates

for $a, b$ for GoF, and for the semi-synthetic procedure for TST. We found that $\hat{a} \approx 49.5$, while $\hat{b} \approx 5.2$, giving a ratio $a/b \approx 10$. The communities, according to $x_{\text{True}}$ are of sizes 636 and 586.

The regime of low $\Lambda$ is explored via sparsification. Fixing a $\rho \in (0, 1]$, sparsification is performed by independently flipping coins for each edge in $G$, and keeping the edge with probability $\rho$. We refer to $\rho$ as the rate of sparsification.

We lastly note that at no sparsification ($\rho = 1$), spectral clustering produces a partition $\hat{x}_1$ such that $d(x_{\text{True}}, \hat{x}_1) = 56$.

**GoF Procedure.**

1. The graph is sparsified at rate $\rho$. Let the sparsened graph be $G_\rho$.

2. For the naïve recovery based scheme, spectral clustering is performed on $G_\rho$ as in the previous section to generate $\hat{x}_\rho$.

3. For the proposed test from §2, the statistic is computed on $G_\rho$.

4. The size of the test is estimated by running the GoF tests with $x_0 = x_{\text{True}}$. For the naïve scheme, we reject if $d(\hat{x}_\rho, x_0) \geq s/2$; for the proposed scheme, we use the test from §2.

5. To compute the power at distortion $s$, we first generate $y$ by randomly inverting the community labels of $s$ nodes in $x_{\text{True}}$. We then run the same procedure as in the previous line, but with $x_0 = y$. Note that the graphs are not edited in any way.

6. The precise implementation details are exactly as in Appendix C.1.1, with the minor difference that we use a regularizer of $\tau = 1$ for spectral clustering.

**TST Proceudre.**

1. Recall that TST requires two graphs as input. The experiment compares the political blogs graph against SBMs.

2. To compute the size, we require a graph with the same underlying communities as $G$. Thus we generate $G'$, which is drawn as an SBM with the underlying partition $x_{\text{True}}$, and parameters $a, b$ as estimated from the political blogs graph $G$.

3. To determine the power of the tests we need a graph with an $s$-far underlying community. For this, we first generate a $y$ such that $d(x_{\text{True}}, y) = s$, as we did in the GoF Procedure. Next, we sample $H$ as an SBM with underlying partition $y$.

4. The graphs $G$, $G'$ and $H$ are now all sparsified at rate $\rho$ to get $G_\rho$, $G'_\rho$ and $H_\rho$.

5. The size of each test is estimated using the TST procedures, as described in Appendix C.1.1 on the pair $(G_\rho, G'_\rho)$. Power is similarly estimated using the TST procedures on the pair $(G_\rho, H_\rho)$.

## C.3 Experiments on the GMRFs

Following the heuristic detailed in §4.3, we naïvely generalise community recovery and testing to this setting, by replacing all instances of the graph adjacency matrix in previous settings with the sample covariance matrix.

The Gaussian Markov Random Field is described by its precision matrix $\Theta$ (i.e., the inverse covariance matrix of the Gaussian random vector on its nodes). We perform a preliminary examination of the possibility of testing changes in communities for an SBM-structured GMRF even when learning the structure is hard or impossible. As described in Section 4.3, we set

$$\Theta = I + \gamma G,$$

where G is the adjacency matrix of an SBM with known parameters. We generate samples from the GMRF as follows:

1. For a fixed number of nodes $n$, we fix an SNR for the SBM, $\Lambda$, and compute $(a, b)$ satisfying this $\Lambda$ so that $b/a = r$.

2. Here, we consider $n = 1000$ nodes and take $\Lambda = 30\Lambda_0$, where $\Lambda_0 = \frac{10}{11}\log(n/100) \approx 2.1$, as before, and $r = 1/10$. We find that $(a, b) \approx (12.34\log n, 1.234\log n)$. Note that since $\Lambda \approx 63 \approx 10\log(n)$, recovery of the communities for a raw SBM at this SNR is trivial.

3. We fix a GMRF parameter $\gamma$. Here, we take $\gamma = 3/(a + b) \approx 0.032$.

4. We can now construct the precision matrix $\Theta$ after sampling $G$ from the SBM. We re-sample to ensure that $\Theta$ is positive-definite, but in practice, for the value of $\gamma$ quoted above, we did not encounter the need to re-sample.

5. To generate i.i.d. samples $\zeta \sim \mathcal{N}(0, \Theta^{-1})$ in a stable manner, we use the following algorithm:

   (a) Compute the lower-triangular Cholesky factor $R$ of $\Theta$, so that $\Theta = RR^{\mathsf{T}}$.
   (b) Sample $\xi \sim \mathcal{N}(0, I)$ from a standard $n$-dimensional multivariate normal distribution.
   (c) Solve for $\zeta$ in $R^{\mathsf{T}}\zeta = \xi$.

   This suffices, since, $\zeta = (R^{\mathsf{T}})^{-1}\xi$ would then have the covariance matrix $(R^{\mathsf{T}})^{-1}R^{-1} = (RR^{\mathsf{T}})^{-1} = \Theta^{-1}$.

6. In this manner, we generate samples from the null and alternate distributions: let $\zeta$, $\zeta'$ and $\upsilon$ respectively denote samples drawn from a GMRF structured using $G$, $G'$ and $H$ respectively. Here, $G$, $G'$ and $H$ exactly are as described in Section C.1.1.

Next, we describe how each of the two schemes is evaluated:

1. Assuming we have $t$ i.i.d. samples of $\zeta$, generated as described above, we estimate the covariance matrix $\hat{\Sigma}$ of $\zeta$ using the standard estimator:

$$\hat{\Sigma} = \frac{1}{t-1}\sum_{i=1}^{t}(\zeta_i - \bar{\zeta})(\zeta_i - \bar{\zeta})^{\mathsf{T}},$$

where $\bar{\zeta} = \frac{1}{t}\sum_{i=1}^{t}\zeta_i$. We then compute the correlation matrix,

$$\hat{C} : \hat{C}_{ij} = \frac{\hat{\Sigma}_{ij}}{\sqrt{\hat{\Sigma}_{ii}\hat{\Sigma}_{jj}}},$$

which will be used in place of the adjacency matrix for both two-sample testing schemes.

2. Similarly, we compute $\hat{C}$, $\hat{C}'$ and $\hat{D}$ from $\zeta$, $\zeta'$ and $\upsilon$ respectively.

3. The naïve two-sample test based on recovery and comparison is evaluated exactly as described in Section C.1.1, except that $\hat{C}$, $\hat{C}'$ and $\hat{D}$ are used in place of $G$, $G'$ and $H$ respectively. False alarm and missed detection rates are also computed in exactly the same way.

4. The two-sample test based on Algorithm 1 has several important variations:

   (a) We use the test statistics

   $$T_{\text{Null}} = T^{\hat{x}}(\hat{C}) - T^{\hat{x}}(\hat{C}')$$
   $$T_{\text{Alt.}} = T^{\hat{x}}(\hat{C}) - T^{\hat{x}}(\hat{D}),$$

   for the null and alternate distributions respectively. Here, $\hat{x}$ has been estimated from $\hat{C}$.

   (b) The threshold for the test is estimated from data. That is, we simulate $M = 100$ samples of $T_{\text{Null}}$ and $T_{\text{Alt.}}$ each, and fit a classifier to differentiate between the two distributions. The classifier used is a simplistic 1-dimensional Linear Discriminant Analysis.

   (c) We estimate false alarm and missed detection rates by applying the classifier to a hold-out dataset. To use the data as efficiently as possible, we use 10-fold repeated, stratified cross-validation, with 10 repetitions.

**Remark on subsampling.**

1. Note that in the two-sample test for GMRFs based on Algorithm 1, we do not subsample $\hat{C}$ as we did before in the case of SBMs.

2. While previously, we had subsampled $G$ to create two subgraphs $G_1$ and $\tilde{G}$ that shared independence properties for ease of theoretical analysis, it should be noted that subsampling results in an effective loss of SNR. This is also the reason why we had to adjust the implementation using a different rate $\eta$.

3. However, it emerges empirically that skipping the subsampling entirely, with a completely dependent $\hat{x}$ and $G$, makes for better separation between the null and alternate distributions, providing a more powerful statistic.

4. Since we could not analytically derive a threshold for this statistic, we presented the subsampled test statistic for the first experiment.

5. Since in the case of GMRFs, we are estimating the threshold from data, we use the more powerful test statistic to show the full extent of possible gains when using a dedicated algorithm for change detection, instead of naïvely looking for changes by learning community structures first.