[Reviews · NeurIPS 2019]

Reviewer 1



Overall I enjoyed reading this paper, and it is of pretty high quality, and moderately original. Even for a non-specialist of hypothesis testing problems (which are classical questions in statistics) the paper is accessible. It is well written, concise and yet clear (if one trusts the proofs, that are in appendix). The hypothesis testing problem studied seems not knew (I'm not a specialist of this literature), but the application to the SBM is. This is not too much "incremental", as theoretical analysis of the SBM is a notoriously difficult problem. The paper provides rigorous bounds, and convincing numerical experiments. My main concern is related to the interest by the ML audience: I wonder how much of a concern is the problem to a pure ML audience, and if that paper would fit better in a good statistics conference. This is not necessary the case. Some detailed comments: _My main concern: There is something that I do not understand and that the authors should clarify (it is actually a crucial point): It is claimed, in particular at line 46, 47 that the case of sparse graphs with bounded degree (bounded in average) is considered, which in the notations of the paper means to me (and in general in the SBM literature) a,b=O(1). But later in the definittio of the SBM it is written Kline 134 that a,b=O(ln n). So they diverge. So formally, even if ln n grows slowly, the graph is not sparse (and for non-sparse case the analysis is much more easy in general). Do I miss something? Even if yes, this point is misleading and should be clarified. _please define the acronyms FA and MD(x) in eq (1) _I think the definition of R_{TST} in (2) is wrong: as you restrict the sup over x,y s.t. d(x,y) >= s, there is an error when the test is different than 1, while you put an indicator 1(x = y), which is 0 all the time as you consider x,y s.t. d(x,y) >= s. _line 179: please specify that you allow sel edges (which is unusual), otherwise the second Bin(n^2/4, a/n) is not correct _lines 183, 184 are very mysterious, please clarify what you mean here _lnie 308: "and" is reputed twice

Reviewer 2



The paper presents a solid contribution into the statistical literature related to the stochastic block model. The related work on the Goodness of Fit and Two Sample Testing is up to my knowledge very well covered. In my opinion the paper present a couple of drawbacks that I list below: I believe that the reason SBM is so widely studied largely because some of the constants, that in other statistical models might be very hard to determine, can be determined in this model. This being the main attraction of the model, and given that the present paper does not specify the constants, this limits the significance or the contribution. The manuscript is not consistent in terms of the scaling of the degree that is considered in various parts. The abstract states "we consider the challenging sparse SBM regime, where degree-per-node is constant". Then for instance the results on [DAM17] are used, but those apply to degree growing with n, analogous results of \Lamba < 2 for constant degrees are due to Mossel, Neeman, and Sly, "Stochastic Block Models and Reconstruction", 2012. Towards the end of page 5 they state: "a, b = O(log n) considered in this paper", contradicting the abstract. Are the obtained result useful in practice? The experimental part contains results in the blog network, but it is also stated that "TST plot suffers since the political blogs graph is not completely described as a 2-community SBM". Real networks are indeed rarely well described by the 2-community SBM, the authors should comment more on this issue. (minor) The papers studied SBM with two equal-sized groups, this should be specified from the beginning as this is a particular case of the more general model. And several of the informally stated claims in the first 3 pages do not apply to the general case. -------- post-feedback: I have read the authors answer and the other reviews. While the feedback clarified some issues (sparsity for instance), it also made it clear that applicability of the theory in its present form is limited. In my opinion the paper remains of borderline originality.

Reviewer 3



The question addressed in this paper is interesting and the results non trivial in my opinion. The mathematical methods seem quite standard - Bernstein inequalities and Le Cam's method. Experiments are also provided to validate the results on real data sets. But I did not look at this part. I have not looked at this part and my feeling is that much more investigation would be needed to really validate the modeling on real datta sets. The paper is clearly written. The introduction gives a detailed review of the literature. Review update: in their answers to the reviewers the authors have essentially clarified technical concerns. The question of the relevance of the theory to experimental situations has come up. I am happy to support the paper even if the applicability of the theory is questionable since there seem to be new serious theoretical results w.r.t the literature. However if the experiments show that the theory has unclear applicability the authors should honestly state it, and underline that the main value of the paper is its theoretical contribution.

[Author Response · NeurIPS 2019]

We thank each of the reviewers for their comments and suggestions.

**Clarification on sparsity regime** (raised by $R_1$ and $R_2$).

In our submission, we used the upper-bound notation $a, b = O(\log n)$ which includes $O(1)$.

We did so because behaviour exhibited in different regimes is fundamentally dichotomous. We wanted to be succinct
in gathering all of the results in different regimes.

As seen from Line 63 for *large changes*, reliable testing is possible in the $O(1)$ sparsity regime - the comments
following Thm. 2 explore this. Nevertheless, for *small changes* (see Line 67, and Thm. 1), it is impossible to reliably
test with $a, b = O(1)$ - rather, $a, b = \Omega(\log n)$ is both necessary and sufficient for testing small changes. Overall, this
is one of the main messages of the paper: the SNR requirements for testing large and small changes are qualitatively
different.

Thus, to present all of our testing results, we must technically permit $a, b$ to vary from constant to logarithmic, even
though we are primarily focused on what is possible when $a, b = O(1)$. That said, we will revise the surrounding text
to provide a bit more context for this technical note.

In addition, our thanks to $R_2$ for bringing to attention our oversight in not citing Mossel, Neeman, and Sly here, which
we will amend.

**Practicality** (raised by $R_2$ and $R_3$). First, we note that our approach can be naturally extended to more practical
settings (e.g., many communities, degree correction). For instance, for multi-community setting, given the number of
communities, one can again recover weakly (e.g., by leveraging SDP methods of Fei and Chen), and then adapt our
statistics in a straightforward manner. However, this will affect the SNR thresholds for testing, and fully characterising
the dependence on the number of communities, etc. is messy and non-trivial, enough to merit further work (mirroring
the development of the recovery literature).

Similarly, while we present preliminary exploration of this in the experiments section, validation of these methods and
models on real-world networks is a non-trivial task. We think that extensive pursuit of these here would distract from
the primary theoretical considerations of the paper.

We thank you each for bringing up these important questions, and we will include a discussion of these as directions
for future research.

**Constants** (raised by $R_2$). Note that, as stated on line 157, each constant in the paper can be *explicitly bounded*,
although these bounds may not be the tightest possible. For instance, in the limit as $n \nearrow \infty$, the TST result for large
changes shows impossibility of reliable testing if $\Lambda < 1$, and also that $\Lambda > 4$ is sufficient for reliable testing. The
lower bound is explicitly discussed in the proofs, see Line 866 of the supplement. The upper bound follows from the
proof of Thm. 2 and the work of Mossel, Neeman, and Sly. Non-asymptotic results are also discussed.

From our perspective such gaps within a constant factor of each other is acceptable. Establishing exact constants in
the SBM can be quite challenging, and, even for recovery, these are only known in very particular settings, such as
exact recovery or weak recovery with distortion $n(1/2 - \varepsilon)$. This is certainly an interesting problem, one that likely
requires a dedicated effort to resolve. We will add a discussion of this question to a future work section.

**Balance** (raised by $R_2$). We will try to highlight the balance assumption in the introduction, although we note the
use of 'balance' in the title precisely for this purpose. (As an aside, since submission we have extended the theorems
to unbalanced but linearly-sized communities, perhaps ameliorating this concern.)

**Lines 183,184** (raised by $R_1$). This was meant to be a brief comment explaining why we switch, based on the
relative sizes of $a$ and $b$, from a test based on counts of edges across communities $N_a^{x_0}(G)$ to one based on counts of
edges within communities $N_w^{x_0}(G)$. Both of these counts are of roughly the same signal strength, but depending on
the regime ($a > b$ vs. $a \leq b$), one count will have a lower noise level. We will clarify this comment so that it reads
more smoothly.

**Typographical errors, and presentation suggestions.** We thank the reviewers for pointing these out, especially $R_1$
for catching one in a definition (the sup should be over $d = 0$ *or* $d \geq s$)! We will correct these.

[Meta-Review · NeurIPS 2019]

This paper generated major post rebuttal discussions. Overall, the paper seems theoretically sound and with interesting ideas for the topic of community detection, and this topic has its place at Neurips. Acceptance is thus recommended.